# Karst spring discharge modeling based on deep learning using spatially distributed input data

Andreas Wunsch[1], Tanja Liesch[1], Guillaume Cinkus[2], Nataša Ravbar[3], Zhao Chen[4], Naomi Mazzilli[5], Hervé Jourde[2], and Nico Goldscheider[1]

[1]Karlsruhe Institute of Technology (KIT), Institute of Applied Geosciences, Hydrogeology, Kaiserstr. 12, 76131 Karlsruhe, Germany

[2]HydroSciences Montpellier (HSM), Université de Montpellier, CNRS, IRD, 34090 Montpellier, France

[3]ZRC SAZU, Karst Research Institute, Titov trg 2, 6230 Postojna, Slovenia

[4]Environmental Resources Management, Siemensstr. 9, 63263 Neu-Isenburg, Germany

[5]UMR 1114 EMMAH (AU-INRAE), Université d'Avignon, 84000 Avignon, France

**Correspondence:** Andreas Wunsch (andreas.wunsch@kit.edu)

**Abstract.** Despite many existing approaches, modeling karst water resources remains challenging as conventional approaches usually heavily rely on distinct system knowledge. Artificial neural networks (ANN), however, require only little prior knowledge to automatically establish an input-output relationship. For ANN modeling in karst, the temporal and spatial data availability is often an important constraint, as usually no or few climate stations are located within or near karst spring catchments. Hence, spatial coverage is often not satisfactory and can result in substantial uncertainties about the true conditions in the catchment, leading to lower model performance. To overcome these problems, we apply convolutional neural networks (CNN) to simulate karst spring discharge and to directly learn from spatially distributed climate input data (combined 2D-1D-CNN). We investigate three karst spring catchments in the Alpine and Mediterranean region with different meteorological-hydrological characteristics and hydrodynamic system properties. We compare the proposed approach both to existing modeling studies in these regions and to own 1D-CNN models that are conventionally trained with climate station input data. Our results show that all models are excellently suited to model karst spring discharge (NSE: 0.73-0.87, KGE: 0.63-0.86) and can compete with the simulation results of existing approaches in the respective areas. The 2D-models show a better fit than the 1D-models in two of three cases and automatically learn to focus on the relevant areas of the input domain. By performing a spatial input sensitivity analysis, we can further show their usefulness to localize the position of karst catchments.

## 1 Introduction

Karst aquifers and karst springs are crucial for freshwater supply in many regions and 9% of the global population partly or fully rely on karst water resources (Stevanović, 2019). Karst systems in general are characterized by high structural heterogeneity due to the at least in large parts unknown conduit network, which controls the highly variable groundwater flow. These factors make modeling difficult, nevertheless different approaches exist, which Jeannin et al. (2021) classify as hydrological models (fully distributed models), pipe flow models (semi-distributed models), and data-driven models (including reservoir models). Artificial neural networks (ANN) or its subgroup of deep learning (DL) models are part of the last group. In contrast to the other

two categories, which usually require detailed system knowledge in order to achieve high quality results, DL approaches offer an alternative possibility of modeling, by being able to establish an input-output relationship automatically, without detailed system knowledge necessary. Even though ANNs are not a standard method in karst modeling yet, different types of ANNs have been applied in modeling karst water resources for quite a long time. As one of the first applications Johannet et al. (1994) showed that karst spring discharge modeling is possible with ANNs. Since then, application of ANNs in hydrology in general received ever growing research attention (e.g. Maier and Dandy, 2000; Maier et al., 2010). This has amplified even more in the last years, mainly because of the recent success of DL models (e.g. Kratzert et al., 2018). Rajaee et al. (2019) more recently reviewed applications of ANNs on groundwater; Sit et al. (2020) summarize applications on hydrology and water resources in general. Recurrent neural networks (RNN), such as long short-term memory (LSTM) (Hochreiter and Schmidhuber, 1997) are standard models for time series modeling, because they possess explicit or implicit memory to remember past time steps, which helps to infer the future. A consequence is that they are trained sequentially, which can make them computationally expensive. Convolutional neural networks (CNN) (LeCun et al., 2015) on the other hand use convolutions along the time axis (1D-CNNs) to learn temporal features and can be trained batch-wise, which therefore usually makes them computationally favorable over RNNs. One example for this fact exists in the related domain of groundwater level forecasting, where Wunsch et al. (2021) showed that 1D-CNNs are considerably faster than RNNs in the case of single site model application. CNNs at the same time exhibited more stable results through less dependency on the random network initialization and achieved some of the highest performances in this specific study (better than LSTM). Other authors similarly applied CNNs successfully for either groundwater level forecasting (Afzaal et al., 2020; Lähivaara et al., 2019; Müller et al., 2020) or rainfall-runoff modeling (Van et al., 2020; Hussain et al., 2020). Müller et al. (2020) find in contrast to Wunsch et al. (2021) that CNNs take a considerably longer time to optimize than LSTMs, yet both studies agree, that they outperform LSTMs in terms of accuracy. Given these favorable properties of CNNs, we choose 1D-CNNs for karst spring discharge modeling for our study. To our best knowledge Jeannin et al. (2021) is the only study yet, applying CNNs for karst spring discharge modeling in some first experiments and they also find CNNs to be superior over LSTMs in terms of testing performance.

Data-driven approaches in general are considered to be black boxes. A way to still build confidence in a model's decisions is to understand what the model is doing (ideally, even why) by using explainable artificial intelligence (XAI). There are different approaches, which are potentially suited for this purpose, depending on the specific goal. Such approaches not only are useful to gain trust, but also help during model building to debug the model and to check what aspects it is focusing on (McGovern et al., 2019). The class of wrapper methods (Kohavi and John, 1997) incorporates both the data and the trained model to interpret what a model has learned. Methods from this class are for example impurity importance for determining feature importance in random forest (RF) models (Louppe et al., 2013), permutation importance (Breiman, 2001) both for RF and DL models, and partial dependence plots (Friedman, 2001) that also reveal why a predictor is important. See McGovern et al. (2019) for an overview on these and several other model interpretation and visualization methods. Especially for image alike data, input sensitivity approaches seem suitable, as focus regions of the model on the image can be visualized. Two well known approaches are occlusion sensitivity (Zeiler and Fergus, 2014) and RISE (Randomized Input Sampling for Explanation) (Petsiuk et al., 2018). Both methods show how relevant each pixel or area is for the decision of the model (image classification)

and can generate an importance heatmap (saliency map) for visualization. The idea behind both algorithms is to use masked versions of an input image and by obtaining the respective model output to learn the focus regions. A very closely related approach to generate a saliency map was recently proposed by Anderson and Radic (2021), which in contrast to RISE and occlusion takes the physical meaning of the absolute value of each variable at each pixel into account during the perturbation of the input data.

One drawback of the 1D-CNN approach, as well as most other data-driven approaches, is the dependency on high data availability and quality. However, climate stations are often not available within the catchment itself, do not match the data availability of the discharge time series (period or temporal resolution), or are more distant and thus do not truly represent the climatic conditions within the catchment. Gridded climate data can provide a solution to such data availability problems. Several openly available products exist (e.g. ERA5-Land (Muñoz Sabater, 2019), E-OBS (Cornes et al., 2018)), which provide climate data for several decades and with, in terms of karst spring modeling, appropriate temporal (hourly or daily) and spatial $(0.1° \times 0.1°)$ resolution. However, especially for karst springs it is not straightforward to extract relevant time series from the gridded data, because the spatial extent of the grid cell containing the location of the spring, usually does not coincide well with the associated spring catchment position. Moreover, especially for karst springs, the catchment is often not well-known, and, for larger springs, can stretch over several grid cells. If the exact position of the catchment is unknown, using gridded data has the advantage that a broader region can be taken into account as input, to let the model learn the relevant grid cells automatically.

Besides such modeling aspects, the delineation of karst catchments is generally important to sustainably exploit but also protect karst water resources by establishing protection zones accordingly. Malard et al. (2015) explain that only few generalizable methods based on models in general for karst spring catchment delineation have been proposed. Instead, delineations usually rely on classical hydrogeological methods such as assessing geology, topography, hydrology, water balance, elaborate tracer tests and geophysical investigations. These methods usually are complex and costly, thus for many karst springs, exact catchment delineations are not available at all or at least contain some uncertainties. Where no information about the catchment is available at all, an approximate localization is advantageous as a first step towards an exact delineation, since it facilitates the application of more elaborate methods like tracer test. There has already been an attempt by Longenecker et al. (2017) to semi-automatically derive approximate catchment boundaries by correlating karst spring discharge events with global precipitation measurement (GPM) gridded data (NASA, 2016). The authors were able to achieve reasonable results with their method, but also noticed that they could not replace conventional methods.

Anderson and Radic (2021) already applied gridded meteorological data to streamflow modeling in western Canada and used a coupled 2D-CNN-LSTM model to directly process spatially distributed input data. They showed that such models learn the relevant parts of the large scale gridded input data for each local or regional streamflow automatically. We adapt and extent this approach to karst spring discharge modeling, however purely based on CNNs by replacing the LSTM part with a 1D-CNN. Similar to the approach of Anderson and Radic (2021), in our proposed methodology the 2D-CNN part learns the spatial features of the input data, while the 1D-CNN part extracts the temporal features, both necessary to simulate the spring discharge time series. With this combined 2D-1D-approach (for the sake of simplicity in the following only 2D-approach)

we can now directly use gridded meteorological data to potentially overcome the common data availability problems when using climate station data for modeling. This approach further does not longer depend on a prior description of the catchment area, other than a very rough estimation of its approximate size to select the gridded data section large enough. Moreover, we investigate the potential of this approach for identifying the approximate catchment location based on a modified spatial input sensitivity analysis from Anderson and Radic (2021). To derive recharge areas based on rainfall-discharge event correlation, as previously done by Longenecker et al. (2017), requires (i) heterogeneous rainfall at catchment scale, (ii) precipitation data with sufficient spatial resolution that capture this heterogeneity and (iii) a karst system without too much dampening of the precipitation signals. These requirements hold for our proposed methodology as well, but a potential advantage of ANNs is their nonlinearity which may better capture the nonlinear relationships between rainfall and discharge.

We explore the applicability of our proposed deep learning approaches with spatially distributed input data in modeling karst spring discharge in three different study areas in Austria (Aubach spring), France (Lez spring) and Slovenia (Unica springs). All three associated karst areas are well studied and for Austria and France, several modeling publications are available as benchmarks. Discharge of Lez spring in France was extensively studied in the past, including several ANN studies. Please refer to Kong A Siou et al. (2011) for an overview about older modeling studies at Lez spring with approaches other than ANN. We omit three newer ANN studies because they either do not focus on modeling discharge (Kong-A-Siou et al., 2015) or train models not on the complete annual cycle (Sep.-Aug. in this region) but on flash-flood events (Darras et al., 2015, 2017). The other ANN studies all use classical multilayer perceptrons (MLP) or recurrent MLPs for discharge modeling and we introduce them shortly in the following. Kong A Siou et al. (2011, 2012) and Kong-A-Siou et al. (2013) use precipitation from three or six gauges, respectively, and all use a similar but slightly varying data basis of 12 to 13 full annual cycles between 1988 and 2006. Testing period is either the single cycle 2002/2003 (Kong A Siou et al., 2012; Kong-A-Siou et al., 2013) or two cycles roughly in the same period (2002-2004)(Kong A Siou et al., 2011). Kong-A-Siou et al. (2014) uses data from 1987 to 2007, however, this time additionally including evapotranspiration and pumping from the Lez aquifer. For Aubach spring in Austria no ANN studies exist, however, other modeling studies are available. Three studies (Chen and Goldscheider, 2014; Chen et al., 2017b, 2018) based on three successive and improved versions of a combined lumped parameter (SWMM) and distributed model, investigate and simulate three springs of this karst system simultaneously. They all achieve high performance in terms of NSE (>0.8), but none of them covered a complete annual cycle as contiguous test period. Additionally, they differ considerably in terms of their individual data basis for modeling (number and position of climate stations used as input data), as well as their testing periods. The shortest test set only had 40 days (in autumn), the longest (Chen et al., 2017b) used one year of data for model calibration and performed a split-sample test on the same data set. This makes a comparison of modeling results among these studies difficult. For the third spring (Slovenia), several earlier modeling studies are available (e.g. Kaufmann et al., 2016; Mayaud et al., 2019; Kaufman et al., 2020; Kovačič et al., 2020), even including ANNs (Sezen et al., 2019), but none of these directly modeled Unica springs discharge, but rather focused on other aspects like cave hydraulics or polje modeling. Besides existing studies we compare the results of the 2D-model with own 1D-CNN models using climate station input data to assess the usefulness and possible advantages of the direct use of spatially distributed input data. As spatially distributed inputs we use either hourly ERA5-Land reanalysis data (Muñoz Sabater, 2019) or daily E-OBS data (Cornes et al., 2018),

depending on the temporal resolution of spring discharge data. We selected these datasets among all openly accessible datasets (e.g. via Copernicus Climate Data Store), because of their available variable set and their spatial and temporal resolution. We introduce them in more detail in the following data section. Finally, we explore the potential of the 2D-approach for karst spring catchment localization by investigating the spatial input sensitivity of the trained CNN models.

## 2 Data and Study Areas

### 2.1 Overview

In this study, we investigate three different karst springs: Aubach spring in the Hochifen-Gottesacker area in Austria (Fig. 1a), springs of Unica river in Slovenia (Fig. 1b) and Lez spring in southern France (Fig. 1c). All springs show different characteristics regarding relevant system properties (e.g. catchment size, complexity of the hydrological system), environmental conditions (e.g. dominant climate, anthropogenic forcing) and data availability (see also Table A1). All areas are well studied and existing data was easily accessible. Further, several previous modeling approaches are available for comparison, except for the Slovenian spring.

### 2.2 Aubach Spring, Austria

Aubach spring is a major karst spring in the Hochifen-Gottesacker karst area in the northern Alps at the border between Germany and Austria. Southern border of the area is the Schwarzwasser valley, which geologically forms the contact zone between the Helvetic Säntis nappe in the north and sedimentary rocks of the Flysch zone in the south (Goldscheider, 2005). In the northern part the dominant karst formation is the Schrattenkalk formation, a cretaceous limestone with a thickness of about 100 m. This Schrattenkalk is structured in folds, which hydrogeologically form parallel sub-catchments (Fig. 1a) that contribute to different proportions to the several springs in the valley (Goldscheider, 2005; Chen and Goldscheider, 2014). In this study we focus on one large, non-permanent spring called Aubach spring (1080 m asl, discharge up to 10 m$^3$/s). The Hochifen-Gottesacker area is largely influenced by seasonal snow accumulation and melting in the elevated regions (>1,600 m asl), which is also clearly reflected in the discharge of Aubach spring by increased baseflow and diurnal snowmelt-induced variations, especially in the months of April to June. Earlier studies by Goldscheider (2005) and Chen and Goldscheider (2014) have identified one major catchment area of Aubach spring with approximately 9 km$^2$ (Fig. 1a), still, to smaller proportions upstream catchments can also contribute to Aubach spring discharge depending on the flow conditions. This applies also to the non-karstified Flysch area directly in the South (southernmost sub-catchment in Fig. 1a), where precipitation events are only relevant during low flow conditions. Then, the surface runoff from this area sinks into an upstream estavelle and contributes via an underground connection to the discharge of Aubach spring. During high flow conditions, the estavelle itself acts as an overflow spring and no contribution from surface runoff at Aubach spring occurs. Generally, the climate in the area can be described as cooltemperate and humid and the mean annual precipitation at the closest used climate station in this study (Walmendinger Horn) is about 2000 mm (2003-2019).

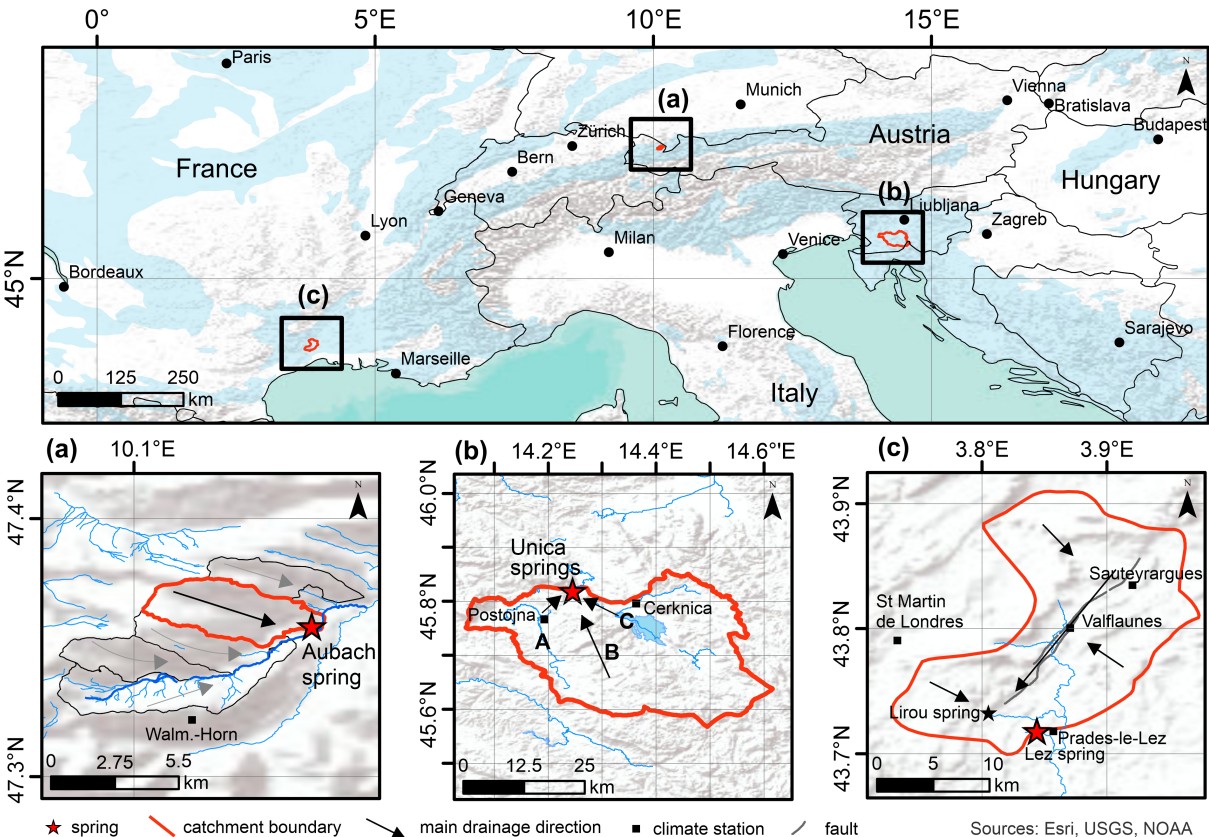

**Figure 1.** Overview of all three study areas, the simulated springs (red star) and their catchments (red lines). Black squares indicate the locations of climate stations used for 1D-modeling (some are outside the shown maps), blue shadings in the upper map show karst areas based on WOKAM (Chen et al., 2017a) (a) Hochifen-Gottesacker karst area and Aubach spring, black lines depict minor contributing sub-catchments; (b) Unica river springs and Javorniki karst plateau (B); (c) Lez spring catchment, Lirou overflow spring (black star) and major fault Corconne-Les Matelles (grey line);

For this study we select Aubach spring because of the good data availability and use 8 years of hourly discharge data provided by the office of the federal state of Vorarlberg, division of water management. Further precipitation and temperature data from three surrounding climate stations are available: Oberstdorf, Walmendinger Horn (shown in Fig. 1a) and Diedamskopf. Additionally, due to the high importance of snow in the area, we run a snowmelt routine as preprocessing of the meteorological input data as described in Chen et al. (2018). This routine is a slightly modified version (after Hock, 1999) of the HBV hydrological model snow routine (e.g. Bergström, 1975, 1995; Kollat et al., 2012; Seibert, 2000), which redistributes the precipitation time series in accordance with probable snow accumulation and snowmelt.

## 2.3 Unica Springs, Slovenia

The Unica springs (450 m asl) are located on the southern edge of a karst polje in SW Slovenia and are important from a biodiversity and water supply perspective. There are two permanent and several temporary springs that feed the Unica river. The joint discharge during 1989-2018 ranged from 1 to 90 m$^3$/s, while the mean discharge was 21 m$^3$/s (ARSO, 2020a). The springs are fed by three clearly distinguishable sub-catchments covering an area of about 820 km$^2$. The main recharge area is the highly karstified Javorniki plateau (up to 1,800 m asl; marked B on Fig. 1b), whose predominant lithology is Cretaceous rocks; mainly limestones, changing in places to dolomites and breccias. To a lesser extent, Jurassic and Palaeogene carbonate rocks are also present. The thickness of the unsaturated zone is estimated to be up to several hundred meters (Petrič et al., 2018, and references therein). To the east, a strike-slip fault zone controls the hydrology of the area, along which a chain of karst poljes developed (between 500 and 700 m asl; marked C on Fig. 1b). Upper Triassic dolomites predominate, changing to Jurassic limestones and dolomites in the south and west, forming aquifers with fracture porosity, which in places have very low to moderate permeability, and in some parts a superficial river network forms. As the karst poljes follow each other in a downward series, they are connected in a common hydrological system with transitions between surface and groundwater flows, and frequent flooding (Mayaud et al., 2019). In the West, the Pivka River Basin (between 500 and 700 m asl; marked A on Fig.1b) consists of poorly permeable Eocene Flysch in the North, which conditions a surface river network. The southern part consists of Cretaceous and Jurassic carbonate rocks forming a shallow karst aquifer. Surface flow occurs during high water levels, receiving additional water from intermittent springs on the western foothills of the Javorniki plateau. The water flow of the sinking rivers in the subsurface from the regions A and C is of the channel flow type. We select the springs for this study because they drain a complex binary karst system of the so-called classical karst, they are well studied with long records of hydro-meteorological data and their hydrology is influenced by substantial snow accumulation and melting. The catchment belongs to the moderate continental climate and is mostly covered with forests. For this study we use daily discharge data from the Unica-Hasberg gauging station (in the following called Unica) (ARSO, 2020a) and daily meteorological data from Postojna and Cerknica climate stations ranging from 1981 to 2018 (ARSO, 2020b). These climate stations (squares in Fig. 1b) are located on the western (Postojna) and eastern (Cerknica) part of the catchment, representing different climate regimes and are separated by the karst massif in between. For Postojna station the following variables are available: precipitation (P), temperature (T), potential evapotranspiration (PET), relative humidity (rH), snow (S) and new snow (nS). For Cerknica station only P, S and nS exist. Average annual precipitation during 1989-2018 is about 1500 mm and on average 33 days of snow cover occur in Postojna (530 m asl) per year, while even longer snow cover is expected on the plateau.

## 2.4 Lez Spring, France

Our third study area is located 15 km north of Montpellier in southern France, within a large and complex karst system delimited by rivers and marly terrains. Eastern and western borders are the Vidourle and Hérault river valleys, northern and southern borders are piezometric limits. At larger scale, northern and southern boundaries are structural boundaries due to Cévennes and Montpellier faults, respectively. The dominant karst formations are Argovian to Kimmeridgian, and Berriasian massive

limestones with 650 m to 1000 m thickness. Infiltration occurs mostly diffuse but also localized through fractures and sinkholes along the basin and through the major geologic fault of Corconne-Les Matelles in the northern part of the basin (indicated by a grey line in Fig. 1c).

The hydrogeological basin associated to the Lez spring has a size of about 240 km$^2$ (Fig. 1c), which is estimated on the basis of the hydrodynamic response to high discharge continuous pumping into the saturated zone of the aquifer (Thiéry and Bérard, 1983). However, the effective recharge catchment of the Lez spring, which corresponds to the extent of Jurassic limestone outcrops, has been estimated to be about 130 km$^2$ (Fleury et al., 2009; Jourde et al., 2014). The Lez karst aquifer is under anthropogenic pressure (i.e. aquifer exploitation for water supply) with pumping performed directly within the karst conduit. The discharge is measured at the spring pool and is regularly null during low water periods, when the pumping rate exceeds the natural spring discharge. Ecological water discharge towards the Lez river (160 L/s then 230 L/s after 2018) is ensured during such periods by a partial deviation of the pumped water to the river. Lirou spring (Fig. 1c) is the main of several overflow springs that activate during high flow periods (Jourde et al., 2014).

The Lez catchment is exposed to a Mediterranean climate, characterized by hot and dry summers, mild winters and wet autumns. Analyses by MeteoFrance show that on average 40% of the annual precipitation occurs between September and November with a high variability across years (Bicalho et al., 2012). The average annual rainfall rate for the 2008-2018 period is 904 mm.

For this study, we use nearly 10 years of daily discharge data provided by SNO KARST (Jourde et al., 2018; SNO KARST, 2021). The temperature data is from the Prades-le-Lez climate station; we use, however, an interpolated precipitation data series that is derived from a weighted average of four rainfall stations (Fig. 1c) (similar to Fleury et al., 2009; Mazzilli et al., 2011), three of them being located on the Lez catchment (Prades-le-Lez, Valflaunès, Sauteyrargues). The fourth station (Saint-Martin-de-Londres) is located few kilometers west of the catchment. Interpolation is in principle possible in this area due to the existing topography; at the same time, interpolation based on Thiessen-polygons (compare Appendix B) also allows compensation for data gaps at single stations. We decided to apply this preprocessing, because all but Saint-Martin-de-Londres climate station show such gaps from time to time, which explains the benefit from including within-catchment precipitation. We do not use pumping data as input in this study, because these were only available for a shorter period of time and such data would also not be available for a real forecast in the future (in contrast to weather and climate data).

## 2.5 Spatial Climate Data

Besides climate station data, we explored raster data from the E-OBS (Cornes et al., 2018), the ERA5-Land (Muñoz Sabater, 2019) and from the RADOLAN (DWD Climate Data Center (CDC), 2020) as spatially distributed model inputs. E-OBS provides daily gridded meteorological data for Europe from 1950 to present, derived from in-situ observations, ERA5-Land provides hourly reanalysis data from 1981 to present. Both are available with a spatial resolution of $0.1° \times 0.1°$ (approx. 8 km $\times$ 11 km for all study areas). Depending on the dataset, different sets of variables are available. In case of E-OBS we initially provide our models with precipitation (P), mean, minimum and maximum temperature (T, Tmin, Tmax), relative humidity (rH) and surface shortwave downwelling radiation (Rad). For ERA5-Land, where a substantially larger set of vari-

ables is available, the following were used as initial inputs: total precipitation (P), 2m temperature (T), total evaporation (E), snowmelt (SMLT), snowfall (SF) and volumetric soil water of all four available layers (SWVL1: 0 - 7 cm, SWVL2: 7 - 28 cm, SWVL3: 28 - 100 cm, SWVL4: 100 - 289 cm). Relevant input variables from both datasets are later selected through Bayesian optimization (see section 3.3). The spatial extent of the input data is chosen very generously for each spring, so that between 6 and 8 additional cells are available as input data around the respective catchments. This prevents a predefinition of the area that needs to be identified as relevant as well as reduces the influence of possible border effects due to the CNN approach using 3x3 filters (compare methodology section). The resolution of ERA5-Land and E-OBS data corresponds to the grid cell size shown in the catchment plots in Figures 1a-c, although each showing a slightly different absolute position of their grid center points. Depending on the temporal resolution of the available spring discharge measurements, we choose the spatial input data in accordance, thus E-OBS for Unica and Lez spring, ERA5-Land for Aubach spring.

Compared to the catchment size of Aubach spring (about $9 \, \text{km}^2$), the spatial resolution (approx. $8 \, \text{km} \times 11 \, \text{km}$) of the gridded input data is extremely coarse. We therefore additionally explore a combination of ERA5-Land input variables (except P) with radar based precipitation data (RADOLAN) that offers a spatial resolution of $1 \, \text{km} \times 1 \, \text{km}$ (DWD Climate Data Center (CDC), 2020). The higher resolved precipitation data from RADOLAN is thus augmented with climate variable values from ERA5-Land (for T, rH, etc.), which were downscaled and re-gridded to match the $1 \times 1 \, \text{km}^2$ RADOLAN grid. Compared to the ERA5-Land section around Aubach spring, for this additional analysis we reduce the spatial extent of the 2D-input data to save calculation time, but still considerably increase the total number of cells due to the higher resolution of the RADOLAN grid.

## 3 Methodology

### 3.1 Modeling Approach

In this study, we simulate karst spring discharge with deep learning models using meteorological input data. As proof of feasibility, we use meteorological data from surrounding climate stations as inputs to 1D-CNN models. However, data from such stations are often limited to precipitation and temperature, rarely more, as well as often exhibit data gaps, and limited record length or coarse sampling intervals. Also, the proximity to the catchment is often not sufficient, which especially in mountainous regions can introduce a distinct error in representing the true conditions within the catchment. This applies especially to variables with higher spatial variability such as precipitation.

Gridded meteorological data can be a solution to these issues, as they usually provide good temporal coverage and sampling intervals, a good spatial resolution as well as a large-scale availability (e.g. continental (E-OBS) or even global (ERA5-Land), see Bandhauer et al. (2021) for a comparison of both products). Further, especially reanalysis data include a larger variable set. When the catchment of the spring is unknown, it remains unclear which cells of the gridded data should be selected to best represent the climate conditions in the catchment, because the actual location of the spring is only a very rough indicator for the location of the catchment. Based on our revised version of the approach of Anderson and Radic (2021), we demonstrate a solution by processing 2D-inputs and letting the model decide automatically, which parts of the input data are relevant to model the spring discharge.

## 3.2 Convolutional Neural Networks (CNN)

Convolutional neural networks (LeCun et al., 2015) are widely applied in several domains such as object recognition (e.g. Cai et al., 2016), image classification (e.g. Li et al., 2014), and signal or natural language processing (e.g. Yin et al., 2017; Kiranyaz et al., 2019). The structure of most CNN models is based on the repetition of blocks that are made up of several layers, typically at least one convolutional layer followed by a pooling layer. The former matches the dimension of the input data (e.g. 2D for image alike data, 1D for sequences such as time series) and uses filters with a fixed size (receptive field) to produce feature maps of the input. The latter performs down-sampling of the produced feature maps and summarizes the features detected in the input. This decreases the total number of parameters of the model and makes it approximately invariant to small translations of the input (Goodfellow et al., 2016). A large variety of model structures based on such blocks, in combination with additional layers in between to prevent exploding gradients (e.g. batch normalization layers (Ioffe and Szegedy, 2015)) or model overfitting (e.g. dropout layers (Srivastava et al., 2014)) are possible; however CNNs usually end with one or several fully connected dense layers to produce a meaningful output.

From earlier studies (Wunsch et al., 2021; Jeannin et al., 2021) we know that 1D-CNNs are fast, reliable and excellently suited for modeling hydrogeological time series, such as groundwater levels or spring discharge. We have shown that they are faster compared to LSTMs, which are often the method of choice for time series modeling, and even outperform them or at least show similar performance (Wunsch et al., 2021). This is in agreement with the findings of (Van et al., 2020) in the domain of rainfall-runoff modeling. In Wunsch et al. (2021) we further show that for the closely related application of groundwater level forecasting, CNNs are less sensitive to the random initialization procedure and thus provide more stable results. Based on these findings we choose CNNs for predicting karst spring discharge in this study and establish two different setups. One setup uses 1D-meteorological input data from surrounding climate stations and applies a 1D-CNN for forecasting. The second approach consistently uses a 1D-CNN to learn temporal features for discharge prediction, but combined with a time-distributed 2D-CNN to learn spatial features directly from gridded climate input data. Compared to the approach in Anderson and Radic (2021) we replace the LSTM by a 1D-CNN to make both setups methodologically consistent. Using CNNs in both setups further helps to assess the influence of using spatially distributed input data in terms of model accuracy, as we do not have to speculate if higher or lower performance might be due to the LSTM model rather than the input data. The general model structure of both setups is shown in Figure 2. They basically use the same 1D-model except the position of the dropout layer. We use Bayesian hyperparameter optimization to select the 1D-filter number, batch-size and input sequence length of each model in both setups.

To reduce the dependency on the random initialization of the models, we use an ensemble with 10 members, each based on a different pseudo-random seed. Further, we implement Monte-Carlo dropout to estimate the model uncertainty from a distribution of 100 results for each of the ten realizations of each model in this study. We derived the 95% confidence interval from these 100 realizations by using 1.96 times the standard deviation of the resulting distribution for each time step. Each uncertainty was propagated while calculating the overall ensemble mean value for final evaluation in the test set. This uncertainty is shown as confidence interval for each of our simulation results in the following. We want to point out, that this uncertainty does not include other sources (such as input data uncertainty) but the random number dependency. All our models

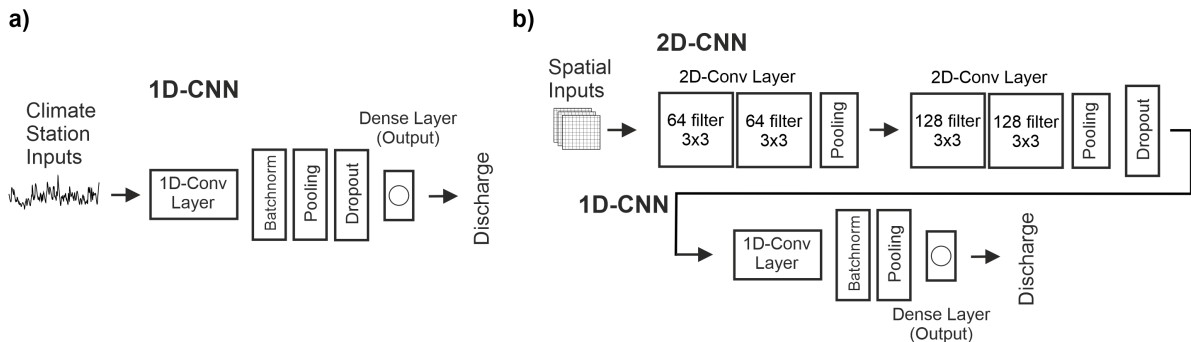

**Figure 2.** Model structures applied for modeling karst spring discharge based on climate station data (a) and gridded meteorological input data (b). Flatten layers are not displayed.

are implemented in Python 3.8 (van Rossum, 1995) and we use the following libraries and frameworks: Numpy (van der Walt et al., 2011), Pandas (McKinney, 2010; Reback et al., 2020), Scikit-Learn (Pedregosa et al., 2011), Unumpy (Lebigot, 2010), Matplotlib (Hunter, 2007), BayesOpt (Nogueira, 2014), TensorFlow and its Keras API (Abadi et al., 2015; Chollet, 2015).

### 3.3 Model Calibration and Evaluation

We split the time series data for each site into four parts according to Table 1. While the first part is used for training, the second part (validation) is simultaneously used to prevent the model from overfitting via early stopping. The model's hyperparameters are optimized according to its performance on the optimization set, while the last set is used as completely independent test set for final evaluation of the model performance without data leakage from training or optimization. Training epoch number

**Table 1.** Data splitting schemes for all study areas (number of values in parentheses).

|  | Time Interval | Training | Validation | Optimization | Testing |
|---|---|---|---|---|---|
| Aubach spring | Hourly | 2012-2017 | 2018 | 2019 | 2020 |
|  |  | (44,807) | (8,760) | (8,760) | (7,320) |
| Unica spring | Daily | 1981-2012 | 2013+2014 | 2015+2016 | 2017+2018 |
|  |  | (11,687) | (730) | (731) | (730) |
| Lez spring | Daily | 2008-2016 | 2017 | 2018 | 2019 |
|  |  | (2,629) | (366) | (365) | (701) |

and early stopping patience are varied manually for each model at each test site. Hyperparameters for the 1D-CNNs of both setups are optimized on the respective optimization set as stated above, maximizing the sum of Nash-Sutcliffe efficiency and $R^2$ (calculated as explained below). The number of optimization steps is also varied manually for each model and is always a trade-off between accuracy and computational costs. In case of many available input variables we treat input variable

selection equally as a global optimization problem and use Bayesian optimization to simultaneously select a proper set of input variables and hyperparameters. Thus, input optimization is used for each 2D-model, as ERA5-Land and E-OBS offer several different climate variables, as well as to the 1D-model of Unica springs, where the climate station records provide additional climate variables such as snow cover. For Lez spring and Aubach spring, only a smaller input variable set is available (mainly precipitation and temperature) and hence fully used. For all models we use an additional input (Tsin), which is a sine curve fitted to the temperature data. This variable can provide the model with noise-free information on seasonality and on the current position in the annual cycle (Kong-A-Siou et al., 2014). Precipitation is the only variable that is not optimized but fixed as input, because it has undoubtedly major influence on the discharge of a karst spring. The optimized hyperparameters, information on some fixed hyperparameters, and a summary of the number of parameters in each model, is given in Appendix Table D1.

We calculate several metrics to evaluate the performance of our models: Nash-Sutcliffe Efficiency (NSE) (Nash and Sutcliffe, 1970), squared Pearson r ($R^2$), root mean squared error (RMSE), Bias (Bias) as well as Kling-Gupta-Efficiency (KGE) (Gupta et al., 2009). For squared Pearson r we use the notation of the coefficient of determination ($R^2$), because we compare the linear fit between simulated and observed discharge, thus of a simple linear model, which makes them equal in this case.

### 3.4 Spatial Input Sensitivity and Catchment Localization

Anderson and Radic (2021) show in their study that combined 2D-CNN-LSTM models can learn to focus on specific areas of the spatially distributed input data and that these make physically sense. We modify this approach and transfer it to karst spring modeling, where we demonstrate that this approach is suited to approximate the location of karst catchments.

We use the Gaussian spatial perturbation approach from Anderson and Radic (2021), which is similar to other input sensitivity algorithms such as occlusion (Zeiler and Fergus, 2014) or RISE (Petsiuk et al., 2018), but in contrary to these methods takes into account the physical meaning of the absolute value of each variable at each pixel during the perturbation. We modify this approach so that only a single input channel (e.g. precipitation) is perturbed at a time for the sensitivity analysis. For details of this approach we refer to the original study. In short it works by perturbing spatial fractions of the input data by adding or subtracting a 2D-Gaussian curve from the input data at a certain location. Both the perturbed and unperturbed data are passed through the trained model to determine the resulting simulation error between them. In this way, after many iterations, heat maps are created that show how sensitive the trained model is to perturbations of certain areas of the input data.

The considered input variables in our study show different properties in terms of spatial heterogeneity and variability. Temperature for example usually exhibits a distinct spatial autocorrelation, meaning that temperature information from outside the catchment area may be used to infer temperature within the catchment area. In contrast, precipitation is less spatially autocorrelated, meaning that precipitation information from outside the catchment area is less related to precipitation from inside the catchment area. Therefore, we hypothesize that the within-catchment precipitation fields will be most important for the model's prediction, and we will test this hypothesis by visually inspecting the sensitivity maps produced by the modified approach of Anderson and Radic (2021). Compared to the original approach by Anderson and Radic (2021), we therefore perturb only single channels at a time, instead of all channels at once, to separate the influence of each channel on the model output.

## 4 Results and Discussion

### 4.1 Aubach Spring

Figure 3a shows the simulation results of the 1D-CNN model for the test period 2020, using only available climate station input data. Error measures indicate a high accuracy of the model simulation: NSE and $R^2$ values both are 0.74, KGE is 0.79. We observe that peaks in winter and spring are underestimated. The snowmelt period, clearly visible by increased baseflow and diurnal variations from April to June, is nicely fitted, as well as the following summer peaks. A short series of discharge peaks in the end of September/beginning of October is not captured. We assume that these were caused by small-scale precipitation events that are not represented in the data of the climate stations used as inputs. Interestingly, diurnal variations, which might be learned during the snowmelt period, are also visible in periods not influenced by snow (e.g. in August). From Chen et al. (2017b) we know the high relevance of snow in this area and by coupling the CNN model with a snow routine data preprocessing, we are able to further improve the model performance (Fig. 3b). We now can achieve a fit with 0.77 for both NSE and $R^2$, KGE increases to 0.84. Our model is able to better fit the second largest peak of the whole dataset, which occurs in February, though, the peak is slightly overestimated, whereas other peaks still tend to be underestimated. The snowmelt period remains well simulated, but shows increasing deviations close the end of the period. The earlier noticed diurnal variations in summer and autumn, now are diminished, which is presumably an effect of the snowmelt preprocessing.

Please note that the 95% model uncertainty from random number dependency, estimated from 10 differently initialized models with a Monte-Carlo dropout distribution from 100 runs each (i.e. 1000 simulations in total), is very low for both modeling results (a+b) compared to the overall variability of the discharge. We assume the spatially limited input data to be the major source of error in the complete modeling procedure, because all climate stations are located outside of the catchment area and thus introduce distinct uncertainty about the true conditions within the catchment. Other modeling approaches (Chen and Goldscheider, 2014; Chen et al., 2017b, 2018) based on combined lumped parameter (SWMM) and distributed models, achieve similar or higher NSE values for the simulation of Aubach spring discharge (0.92, 0.83, 0.80 respectively). As mentioned, the results are, however, hardly comparable with each other and neither with this study. Reasons are (i) different input data (number and position of climate stations), (ii) different simulation periods, and (iii) very different test set lengths. One reason for the slightly lower performance of our model could be that none of the previous studies covered a complete annual cycle as contiguous test period, including high peaks in late winter and strong snowmelt influence in spring and early summer.

Figure 3c shows the results of the 2D-modeling setup using (only) ERA5-land input data. Based on the described optimization procedure, the model uses the following inputs: P, T, E, SMLT, SWVL2 and SWVL4 (for a comparison of selected input variables with other study areas see also Table A1). The performance of the 2D-model is similar to that of the 1D-models, showing a NSE (0.76) and RMSE in-between both, a larger $R^2$ (0.8) but a lower KGE (0.71). This performance is still high considering that the major catchment is extremely small (about $9 \, \mathrm{km}^2$) compared to one ERA5-Land grid cell, and that a large grid section of $14 \times 14$ ERA5-Land cells ($1.4° \times 1.4°$) was used as input. We see that the major peak in February is slightly underestimated, as well as the beginning of the snowmelt period in April; however, the end of this period in May/June has improved now compared to (b). Both 1D-models are superior in estimating the peaks especially during summer, except the

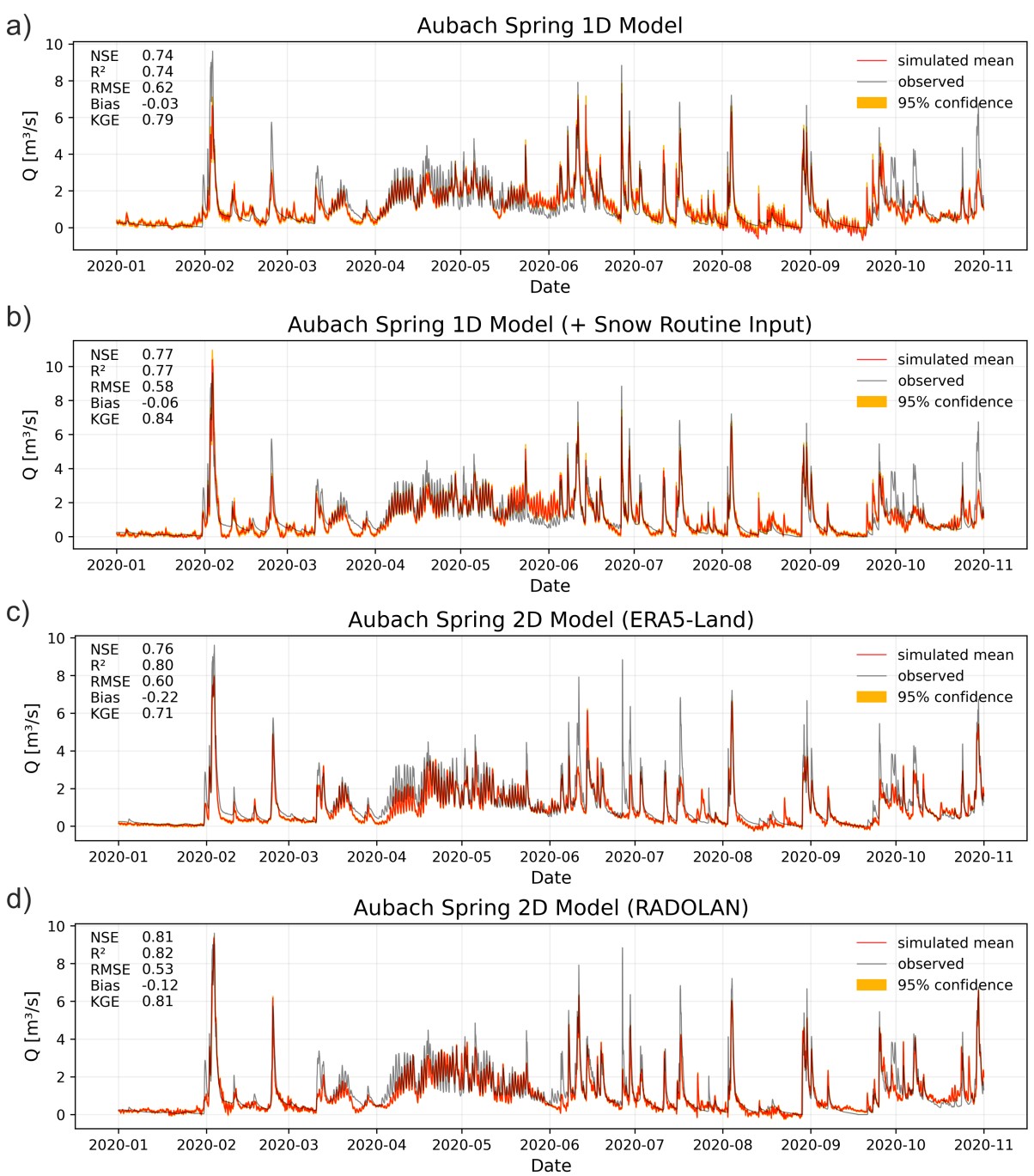

**Figure 3.** Simulation results for the year 2020 at Aubach spring: (a) 1D-model based on climate station inputs, (b) 1D-model with additional snow routine preprocessing, (c) 2D-model based on ERA5-Land gridded data and (d) 2D-model with combination of ERA5-Land data and RADOLAN precipitation input.

already mentioned peaks in September/October, which have improved using the 2D-input data. This supports the assumption that the climate stations do not represent these precipitation events, but the 2D-data does due to its spatial nature.

To account for the small area of the catchment of Aubach spring, Figure 3d shows the results of the 2D-input data, using the spatially higher resolved RADOLAN precipitation data in combination with downscaled ERA5-Land data. We have reduced the spatial extent of the 2D-input, but still have a reasonable buffer around the catchments and, compared to the former 2D-model, increase the grid cell number considerably ($22 \times 22$ or $22^2$ km$^2$). The optimized model uses P, T, Tsin, SMLT, SF, SWVL1/2/4 as inputs, thus omits E and SWVL3. This model shows the best performance of all four models by reaching a NSE of 0.81, R$^2$ of 0.82 and KGE of 0.81. Similar to the model in (c), the beginning of the snowmelt period in April remains slightly underestimated and compared to the 1D-models, the peaks in summer are less well fitted. Nevertheless, we generally see an accurate fit, especially the largest peak in February is well reproduced. Compared to the 1D-approach, the main source of uncertainty for both 2D-models should be the uncertainty of ERA5-Land variables. Their values originate from large grid cells in comparison to the catchment size, thus it is not clear how well they represent the true conditions on catchment scale. A more elaborated downscaling of ERA5 data or other high resolved climate data for a combination with RADOLAN precipitation data might be a promising approach for simulating small catchments like this one. Model uncertainty derived from random number effects and Monte Carlo dropout is (equally to the 1D-models) satisfyingly small. In total, we think that both the 1D and the 2D-approach for this catchment bear substantial shortcomings in terms of how well the input data represents the true conditions in the catchment, even though the simulation results are generally very accurate. On the one hand the climate stations represent the true observed climate, on the other hand this is true only for a very specific point, which is in this case outside the catchment, and embedded into a highly variable topography. The 2D-data have a too coarse spatial resolution compared to the size of the Aubach spring catchment and are themselves modeled (in case of ERA5-Land). We therefore do not think that one approach is superior for this study area, but we can show that even in this case with relatively coarsely gridded input data compared to the catchment size, the 2D-approach offers a decent alternative in case of missing climate station data.

## 4.2 Unica Springs

Figure 4 summarizes the 1D- and 2D-model performance on the years 2017 and 2018 for Unica springs in Slovenia. The simulation of this quite large catchment area (820 km$^2$) is based on the data of only two climate stations (Postojna and Cerknica). All available input variables from both stations except relative humidity from Postojna station and new snow from Cerknica station were used as inputs as selected by the Bayesian optimization model. The 1D-model shows good performance overall (NSE: 0.73, R$^2$: 0.79, KGE: 0.63), including a response for all major discharge events. However, recession slopes especially in 2017 are underestimated substantially and the plateau shapes of the large peaks (e.g. January 2018) are not well captured, but rather simulated as multiple peaks. In general, many of the high flow events at this gauge have a quite long duration of days to even weeks resulting in such plateau-like shapes. This is due to the regular flooding of the polje. After the drainage areas of the polje are completely flooded, there is a progressive back-flooding and a steady rise in the water level, which makes it impossible to accurately monitor the flow conditions under these conditions. Therefore, during the plateau-like peaks, when we cannot observe the true flow; the peaks simulated by the ANN might be conceptually true, which is however not possible

to evaluate. The peak in April 2018 is quite clearly underestimated, whereas the following low flow period (summer 2018) is slightly overestimated. Such overestimation might be due to small scale meteorological events that are detected by the climate stations, but do not well represent the conditions in the whole catchment area. It is also important to notice that during 2014 and 2018 substantial environmental changes occurred in the catchment (Kovačič et al., 2020). During this period a considerable amount of vegetation was destroyed by a series of large-scale forest disturbances. We expect the evapotranspiration changed due to changes in canopy interception, water use, and soil moisture. As a result, spring behavior has likely changed, because vegetation cover is an important element of the water balance and recharge events may have resulted in higher infiltration rates and more intense spring response, as well as more pronounced droughts. The effect of this environmental change on the model performance is hard to evaluate, because it is not part of the training data. However, the model was optimized and validated (early stopping) on a part of the period with environmental changes, which means that the model may infer some information on the changes from these periods (2014-2016). It is not expedient to exclude this change from model building, since this would require to shorten the time series to the period after 2018, thus loosing almost the complete data basis. Due to highly complex hydraulic behavior in this study area, which is for example related to already mentioned polje floodings and to a strongly variable water level in the system that varies also the catchment area, extracting the highly non-linear precipitation-discharge is especially challenging. We generally observe less dynamics in terms of the number of flood pulse events compared to Aubach spring. In terms of intensity of hydrologic variability, discharge rates can vary by about two orders of magnitude. This is primarily due to the large size of the catchment area, the very high degree of karstification of the carbonate rocks, and the fact that the main spring may acts as an overflow spring.

By using the 2D-input data from $18 \times 21$ E-OBS grid cells we were able to improve the model performance substantially (Fig. 4b), reaching now a NSE of 0.83, a $R^2$ of 0.84 and a KGE of 0.80. Model input variables are: P, Tmax, rH and Rad. We generally observe a similar shape of the simulation as for the 1D-model but with overall reduced errors. Still, the plateau shapes of some peaks are not well captured, but the same conceptual understanding as for the 1D-model seems to be learned, which means the model mainly simulates peaks instead of plateau-shaped high-flow events. The slope of the recessions are still generally underestimated, especially the simulation of low flow periods and minor discharge events improve clearly though. The improved results are plausible, because we can expect precipitation events to be represented more accurately in the gridded data that in the point data of only two climate stations, especially considering the size of the catchment. As for Aubach spring, both models show a comparably low model uncertainty based on random number variation and Monte-Carlo dropout, the model uncertainty of the 2D-simulation is even a bit lower than for the 1D-model. Again, we assume the spatially limited climate station data to be the main source of data uncertainty in the 1D-model, because meteorological stations are located on the western and eastern side of the karst massif. The massif itself represents the orographic barrier with different temperature and precipitation regimes that are certainly not captured by the considered meteorological stations. Concerning the 2D data, the grid resolution is sufficiently high to adequately represent the climatic conditions in this large size catchment.

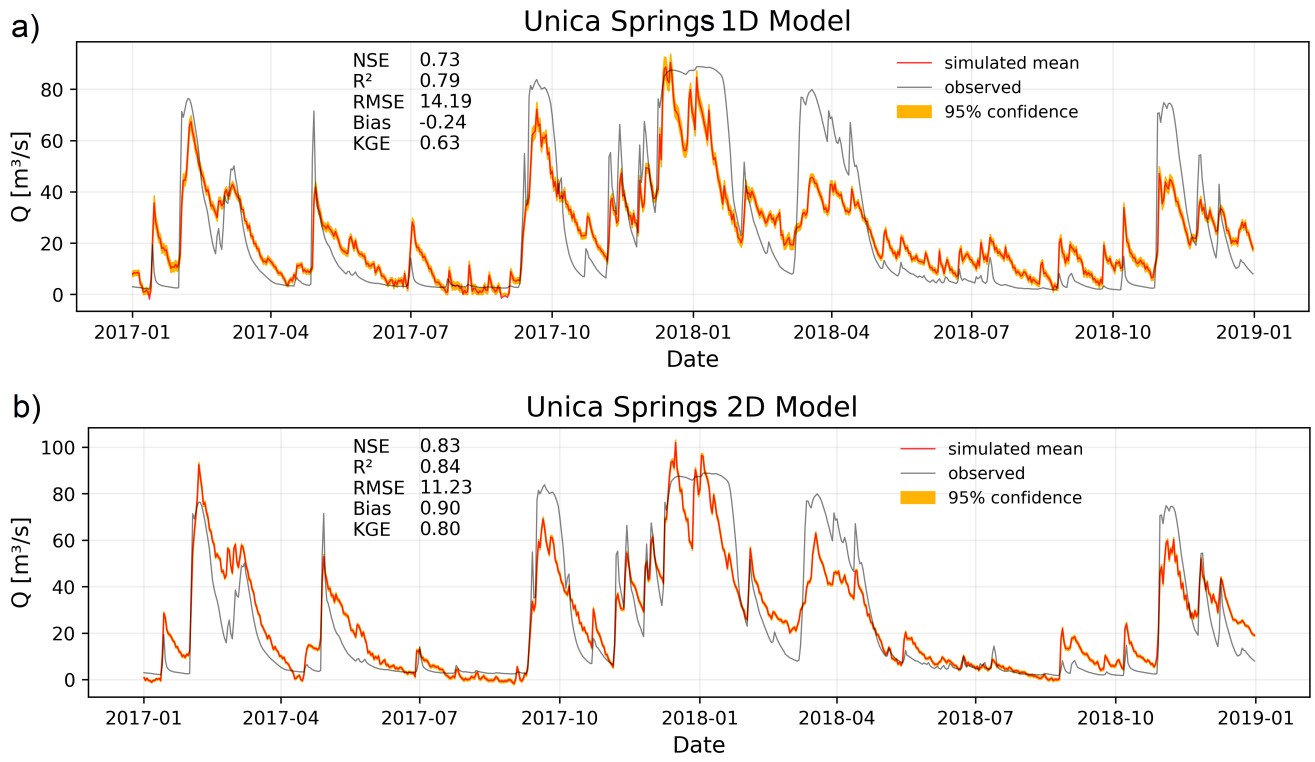

**Figure 4.** Simulation results for 2017-2018 at Unica springs in Slovenia using climate station input data (a) and E-OBS gridded data (b).

### 4.3 Lez Spring

Lez spring represents a third class of study area, as the catchment size (around $240\,\text{km}^2$) is somewhere in between the two others, the climate is Mediterranean and the spring runs dry for a considerable amount of time during the annual cycle due to a constant exploitation of the karst aquifer through pumping. Figure 5 shows both the results for the 1D- (a) and the 2D-model (b). Despite comparably short training (daily data, starting in 2008) we observe a very high fit of the 1D-model above 0.86 for NSE, $R^2$ and KGE. As well the timing of the peaks, the absolute height of the peaks, as the dry periods are simulated

accurately, except some deviations in early 2019.

For the 2D-model we use input data from $19 \times 18$ E-OBS grid cells and the Bayesian model selects only rH and Rad as inputs besides the fixed input P. Considering the high relevance of potential evapotranspiration (PET) in the Mediterranean, it is a bit surprising that temperature, as a major driver of PET, is not selected (neither T, Tmin nor Tmax). However, relative humidity is also important to calculate PET (King et al., 2015) (e.g. low rH favors high evaporation) and the information on

seasonality well encoded in a temperature time series, is presumably deducible from the radiation data (higher in summer than in winter) . The performance of the model is very good, but clearly lower compared to the 1D-model, showing NSE, $R^2$ and KGE between 0.75 and 0.78. Generally, the simulation is better in 2018 than in 2019, which is, however, also a tendency of

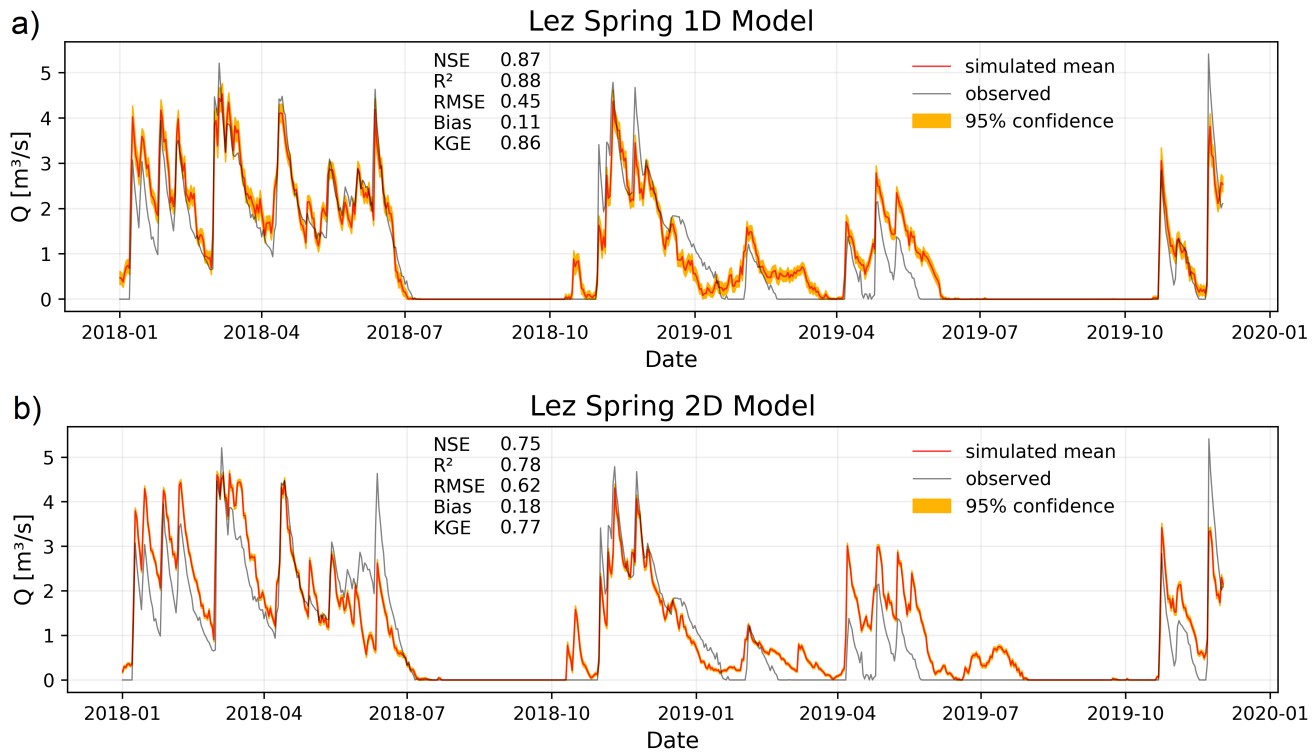

**Figure 5.** Simulation results for 2018-2019 at Lez spring in France using climate station input data (a) and E-OBS gridded data (b).

the 1D-model. The model simulated some non-existent peaks in the dry sections, after all one of them (in Oct. 2018) clearly occurs also in the 1D-model's simulation. Presumably, the input data is accountable for the general performance differences
between both modeling approaches. The climate stations, from which the interpolated precipitation time series is derived, are mainly located inside the catchment and additionally represent a good spatial coverage. Compared to both other study areas, the 1D-input data here best represents the climatic conditions within the catchment. Based on the lower performance of the 2D-model, we conclude that it seems to be harder to extract the relevant relationships between climate forcing and spring discharge from the gridded data. This may be related to a less favorable ratio of grid cell size to the catchment size, than for
the Unica catchment for example. The model uncertainty based on initializations and derived from Monte-Carlo dropout again is small for both model setups, especially during dry periods.

     The results of our models (1D-NSE: 0.87, 2D-NSE:0.75) can compete with the results from several earlier studies (NSE: 0.76-0.88 (Kong A Siou et al., 2011), NSE: 0.76-079 (Kong-A-Siou et al., 2014)), however, we do not beat the maximum performance reported by Kong A Siou et al. (2012) (NSE: 0.69-0.95) and Kong-A-Siou et al. (2013) (NSE: 0.96). Generally,
all studies, including ours, achieve high performance and it is hard to conclude reasons for the superiority of one or other study, as several factors differ among them, such as model types, training and testing periods, or set of input variables. For our study,

we chose not to include pumping data (as used in (Kong-A-Siou et al., 2014)) due to the data availability reasons elaborated in section 2.4, as well as to be consistent in the 2D-modeling approach, which would need an update of the model structure due the 1D-time series character of the pumping data. The 2D-approach still shows very good performance in general, however, in comparison among all mentioned NSE values its performance is rather low. Nevertheless, we conclude if no climate station data would be available to apply a 1D-model, the 2D-approach still offers a reasonable substitute.

## 4.4 Spatial Input Sensitivity Results

The most important results of the spatial input sensitivity analysis from all catchments are shown in Figure 6. In case of Aubach spring modeled with ERA5-Land data (Fig. 6a), we can see that the catchment is smaller than one grid cell. Hence, despite the quite good discharge modeling, we see no clear spatial meaning of the precipitation channel heatmap. We also find a border effect with an almost uniform decrease in sensitivity toward the edges, which is an important reason to choose the spatial extent of the data large enough. This effect could be related to the size of the filter in the convolutional layer ($3 \times 3$), as it sometimes only occurs in the one or two outermost pixels (e.g. Fig. 6c). In combination with zero-padding, which we apply to improve the informative value of the edges and to maintain the data size throughout the convolutions, this may result in such error halo, as also illustrated by Innamorati et al. (2020). Yet its origin remains unclear and not all heatmaps show this pattern (Fig. 6d), which questions the hypothesis of being a purely technical issue. For Aubach spring, precipitation shows only the fourth highest sensitivity (S) in terms of absolute values, while the second most sensitive variable is snowmelt (SMLT), which shows also the best spatial agreement with the catchment area. This is plausible insofar as the discharge for a large part of the time is dominated by snowmelt and to a lesser extent directly by precipitation. We conclude that even though the modeling results are satisfying, not much meaning can be extracted from the spatial sensitivity analysis for such a small catchment, given the existing spatial resolution of the gridded data. Please find heatmaps of all other variables in Appendix Figure C1. The combined approach of RADOLAN and ERA5-Land data (Fig. 6b) shows the heatmap in more detail in relation to the size of the catchment. We show only the precipitation heatmap, because it is the only variable with a native resolution of $1 \, \text{km} \times 1 \, \text{km}$ and we do not consider the spatial patterns of the remaining ERA5-Land-based variables to be meaningful to interpret. We observe that the most sensitive cells are identified close to the spring and at the border between the main catchment and the southern adjacent subcatchment. Due to the small scale of the spatial extent shown in Fig. 6b in relation to the spatial extent of precipitation events, the model is not able to sharply distinguish between precipitation inside and outside the catchment. This is presumably also related to the data, as precipitation is not directly measured, but estimated from radar signals and subsequently adjusted according to measured values from nearby climate stations. It remains unclear if precipitation is spatially resolved with sufficient accuracy in such alpine valleys on km-scale. No plausible reasoning exists for the two separate sensitive areas in the SW and NE corners, however, they are less sensitive then the center cells of the map.

Heatmaps of all four selected E-OBS variables at Unica catchment are shown in Figure 6c. In accordance with our expectation for karst areas, we see the highest sensitivity for precipitation, which visually also identifies the catchment area very well. Especially Tmax and rH show high sensitivities on larger areas, however they are usually highly spatially autocorrelated and do not show a strong spatial heterogeneity like precipitation, which makes it plausible that the model learns from larger areas

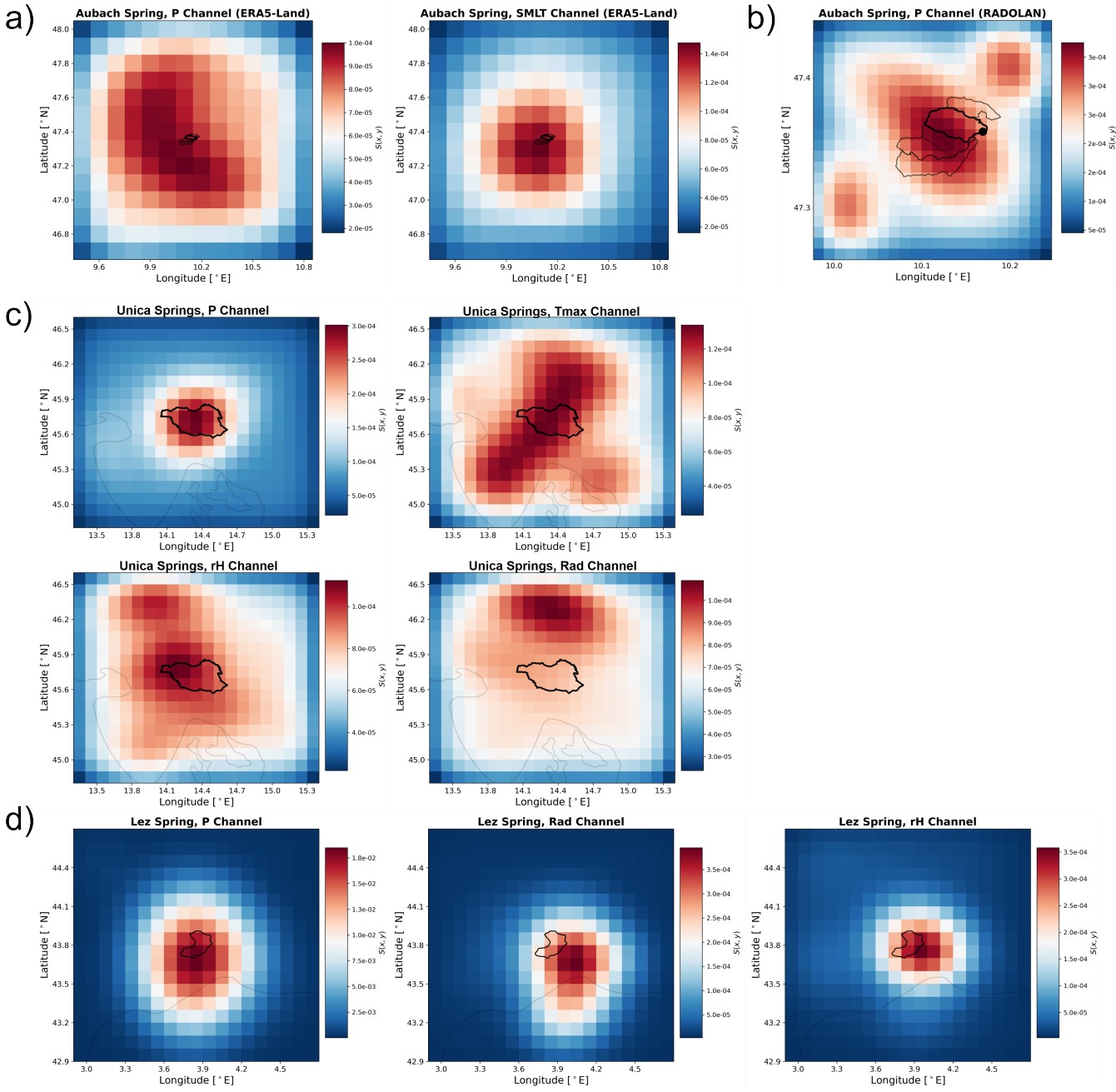

**Figure 6.** Heatmaps of spatial input sensitivity for Aubach spring based on ERA5-Land gridded data (a), for Aubach spring based on RADOLAN precipitation data (b), Unica springs (c) and Lez spring (d) both based on E-OBS gridded data. In case of (c) and (d), light-grey lines indicate the coastlines for orientation.

and does not concentrate strongly on the catchment itself. The model further identifies an area in the north as most sensitive for radiation.

Heatmaps of the 2D-Lez spring model are shown in Figure 6d. In this area the model very strongly ignores large parts of the input data (dark blue, no visible border effects) and comparably sharply identifies the relevant area for the spring. This might be related to the higher spatial heterogeneity of precipitation in Mediterranean climate (Fresnay et al., 2012), which in this specific region has a special importance (severe flash floods known as Cévenol episodes (Kong A Siou et al., 2011)). Generally, we observe a slight south and east shift of the highest sensitivity compared to the catchment position. This might be related to the performance of the 2D-approach, which could not compete with the 1D-models. Maybe the model did not exactly learn the most relevant spatial features. The most sensitive variable is precipitation, while the rH channel shows the best spatial fit. We furthermore see that the size of the catchment is about the minimum size to produce meaningful heatmaps based on this given grid resolution, which corresponds also to our interpretation of the 2D-model performance shortcomings in comparison with the 1D-approach.

Given the spatial resolution of the used input data, the obtained heatmaps, and the simulation results of all three catchments, the Unica springs catchment seems to be most appropriate to further investigate the usefulness for catchment localization. It has the highest ratio of catchment size to data resolution and exhibits both generally high performance of the ANN models, and especially a considerably improved performance when using spatially distributed inputs compared to climate station input data. Thus, we used the Unica springs to conduct additional experiments to investigate the sensitivity of our approach to the absolute catchment location within the considered area of the input data. Figure 7 shows the results of these experiments, where we shifted the 2D-input data boundaries in such a way that the catchment is located in one of the four corners or edges, leading to eight additional modeling results, named by the position of the catchment in the considered area of the input data. (e.g. *upleft*: catchment in the upper left corner). First of all, we find that all models successfully model the spring discharge curve and similarly learn the relevant grid cells of the considered input area, i.e. they are able to learn the approximate position of the catchment. The NSE values vary moderately between 0.80 and 0.85 among all models. The heatmaps of the precipitation input channel visually well identify the location of the catchment for each of the different considered areas of the input data. We find that regardless of the catchment's position within the considered areas of the input data, the resulting high-sensitive area in the P channel well indicates the true catchment position. For the heatmaps of the other input channels, we see that usually larger areas are identified as relevant and more variations between the models occur. Two things are particularly noticeable here. First, the identified sensitive input areas are generally slightly smaller for the *up\** models, which is possibly related to the fact that the considered area of the input data is shifted towards the Mediterranean Sea, where no input data are available in the E-OBS dataset (compare the grey coastline). These areas contain zeros or mean values and show no temporal variation that could be used to model the spring discharge. Second, the noticeably best performing model (*downleft*, NSE of 0.85) is the model with the least fraction of no-data cells (due to the Sea). Intuitively, we would not have expected the best performance here, but rather with the *upright* model, since there it is almost predetermined where the model has to learn. So the model seems to be able to use the larger amount of "useful data", even outside the catchment, to improve the overall performance. To possibly delineate a catchment from these results, a strategy has to be developed regarding the sensitivity contrast between the

catchment and its surroundings. From our results we conclude that focusing on the precipitation channel is the most promising approach for potential catchment delineation. This makes, however, only sense if (i) precipitation is sufficiently heterogeneous at the scale of investigation, (ii) if conceptually spring discharge is mainly driven by precipitation (not snowmelt for example) and (iii) the gridded climate data is provided in a relatively high spatial resolution compared to the catchment size. Please find the precipitation channel heatmaps for Aubach spring and Lez spring in Appendix Figures C2 and C3.

In summary, we observe that the approach in its current form can produce meaningful heatmaps for at least roughly locating karst spring catchments. At least for the precipitation channel, we showed that the location of the catchment is successfully learned, regardless of the position within the considered area of the input data, if the ratio of catchment size to grid cell size is favorable (as for Unica springs). We notice that it generally works better the larger the catchment area, especially in relation to the grid cell size, but the absolute size of the catchment itself appears to be also important. For small catchments it seems harder to extract precise catchment locations, even if spatially finer-resolved data are available. This might be related to the fact that at small scales, even precipitation has a distinct spatial correlation, which can lead to higher sensitivity also in areas outside the catchment. However, one should keep in mind that these conclusions are only tendencies as we only investigated a small number of catchments. To develop a catchment delineation strategy, future investigations should analyze more catchments with adequate ratio of size to grid cell resolution, such as Unica catchment. Moreover, it can be expected that more and better gridded meteorological data products will be available in the future, which might lead to better results with the proposed methodology also for catchments with varying sizes.

## 5 Conclusions

From the obtained insights we can conclude that karst spring discharge can be predicted accurately with the presented 1D and 2D-approaches. Their performance competes with that of existing models in the three study areas. One main advantage compared to conventional modeling approaches in karst is that in order to obtain precise discharge simulations, far less prior knowledge of the system under consideration is required. Thus, using ANNs can generally reduce the amount of preliminary work that would be required to gain such sufficient system knowledge. We can further show that gridded climate data can provide an excellent substitute for non-existent or patchy climate station data. This does not require knowledge of the exact catchment area, which is a critical component especially for karst springs. Rather, coupled 2D-1D-CNNs can be used to generate a first approximation of the catchment location. However, as it was shown, this approach still needs further development to more accurately localize the catchment, for example by modifying the input sensitivity approach and by defining a routine to infer the catchment location from the sensitivity data, other than visual inspection. An important factor to achieve more accurate catchment localization is 2D-meteorological input data with a finer spatial resolution in relation to the catchment size, because we found the approach to work best for the largest catchment. Additionally, a sufficient heterogeneity of precipitation in comparison to the catchment size is necessary, which, however, cannot be controlled, but possibly limits the application in some karst areas. Given these developments and conditions, the approach's capabilities to delineate karst catchments should be further investigated, ideally including an evaluation against tracer tests and hydrogeological studies. In terms of accuracy, we

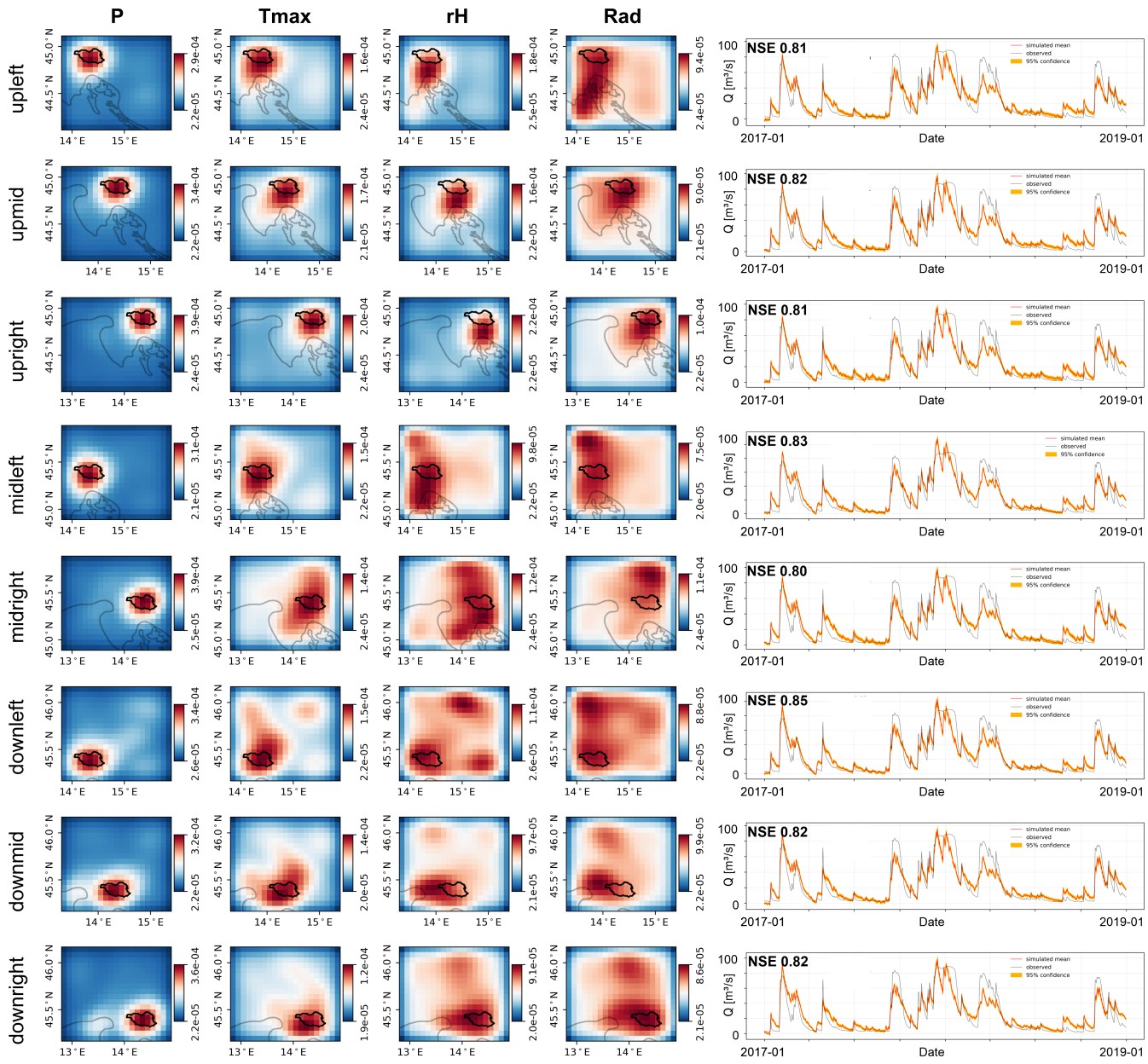

**Figure 7.** Heatmaps of spatial input sensitivity for Unica springs catchment based on E-OBS gridded data. The considered area of the gridded input data is shifted to demonstrate the spatial learning capabilities of the models.

do not find that one of the tested model setups (1D and 2D) is fundamentally superior. A key benefit of the 2D-approach, which uses spatially discretized input data, is the spatially and temporally complete nature of the data and the number of variables available for study. Furthermore, for many areas the openly available 2D-climate data are easier accessible than climate station data, which still have to be collected from various different authorities, if accessible or existing at all. A weak spot of the 2D-approach is a substantially higher computational effort due to the large number of model parameters and the larger amount of data that has to be processed during training and optimization. In summary, gridded meteorological data is useful to overcome missing climate station data and to get a quite good idea of the spatial extent of larger catchments, given sufficiently small grid cell sizes.

*Code and data availability.* We provide complete model codes on Github (AndreasWunsch/CNN_KarstSpringModeling) (Wunsch, 2021). Due to redistribution restrictions from several parties we cannot provide a dataset. Nevertheless, the data is available from the respective local authorities listed in the main text and in the following. 2D-datasets (E-OBS, ERA5-Land) are fully accesible online via Copernicus (cds.climate.copernicus.eu). Aubach spring discharge and climate data from surrounding climate stations in Austria are availble on request from the office of the federal state of Vorarlberg, division of water management, Oberstdorf station data (German Meteorological Service) is available online (opendata.dwd.de). Data from Slovenia can be retrieved from ARSO (Slovenian Environment Agency)(ARSO, 2020a, b). Lez spring discharge was provided by SNO KARST (2021), climate data is available on request from MeteoFrance.

## Appendix A: Study Area Comparison Table

**Table A1.** Summary and comparison of different aspects of all three study areas.

| | Aubach Spring | Unica Springs | Lez Springs |
|---|---|---|---|
| Country | Austria | Slovenia | France |
| Climate | cooltemperate and humid | moderate continental | mediterranean |
| Catchment Area [km$^2$] | 9 | 820 | 240 |
| mean Precipitation [mm/year] | 2000 | 1500 | 904 |
| Station, Period | (Walm.-Horn, 2003-2019) | (1989-2018) | (2008-2018) |
| spatially distributed input datasets | ERA5-Land, RADOLAN | E-OBS | E-OBS |
| Offered variables | P, T, Tsin, E, SMLT, SF, SWVL1-4 | P, T, Tmin, Tmax, Tsin, rH, Rad | P, T, Tmin, Tmax, Tsin, rH, Rad |
| Selected variables | ERA5-Model: P, T, E, SMLT, SWVL2, 4 RADOLAN-Model: P, T, Tsin, SMLT, SF, SWVL1, 2, 4 | P, Tmax, rH, Rad | P, rH, Rad |
| Omitted variables | ERA5-Model: Tsin, SF, SWVL1, 3 RADOLAN-Model: E, SWVL3 | T, Tmin, Tsin | T, Tmin, Tmax, Tsin |

## Appendix B: Lez Catchment Precipitation Interpolation

The Thiessen's polygon interpolation method consists of calculating a weighted average of the precipitation data by allocating a contribution percentage to each meteorological station, based on its influence area on the catchment. These influence areas are calculated through geometric operations. First, we draw straight-line segments between each adjacent station, then we add the perpendicular bisectors of each segment, which will define the edges of the polygons. Each meteorological station thus corresponds to a particular polygon, for which the precipitation over the surface is assumed to be the same as the measured precipitation at the station.

The weighted average of the precipitation $P_{wa}$ at each time step is calculated as follows:

$$P_{wa} = \frac{\sum_{i=1}^{n} A_i P_i}{A} \tag{B1}$$

With $n$ the number of meteorological stations, $A_i$ the area (over the catchment) of the polygon corresponding to the $ith$ station, $P_i$ the precipitation measured at the $ith$ station and $A$ the area of the catchment.

# Appendix C: Heatmaps

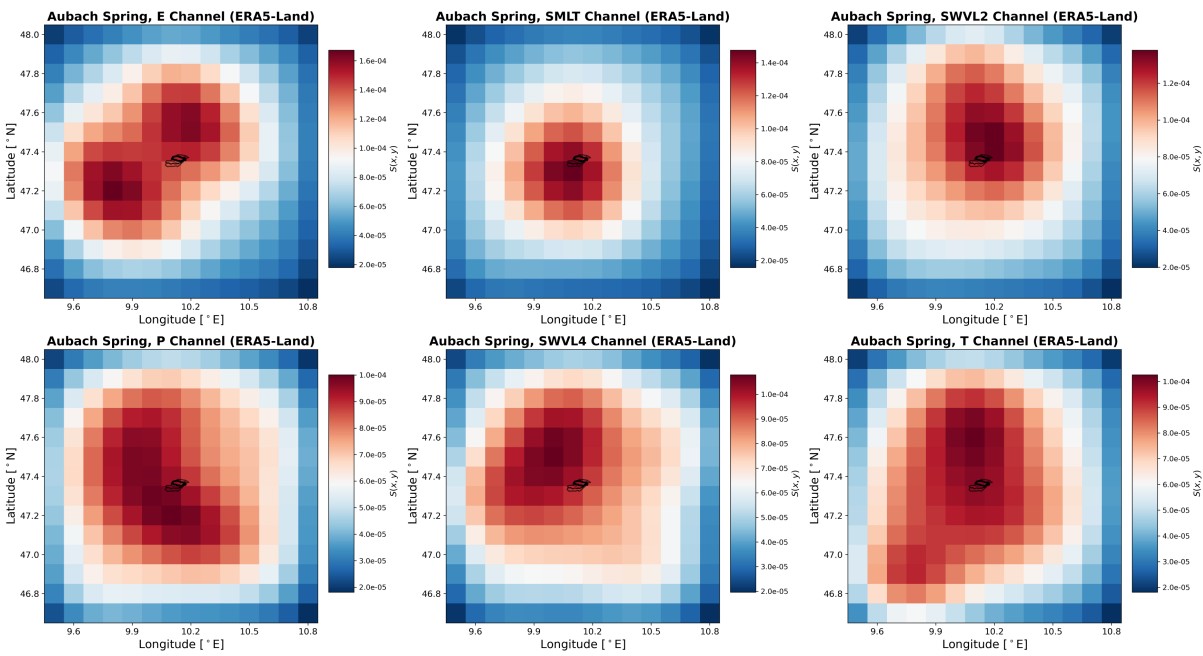

**Figure C1.** Spatial input sensitivity heatmaps for Aubach spring based on ERA5-Land gridded data.

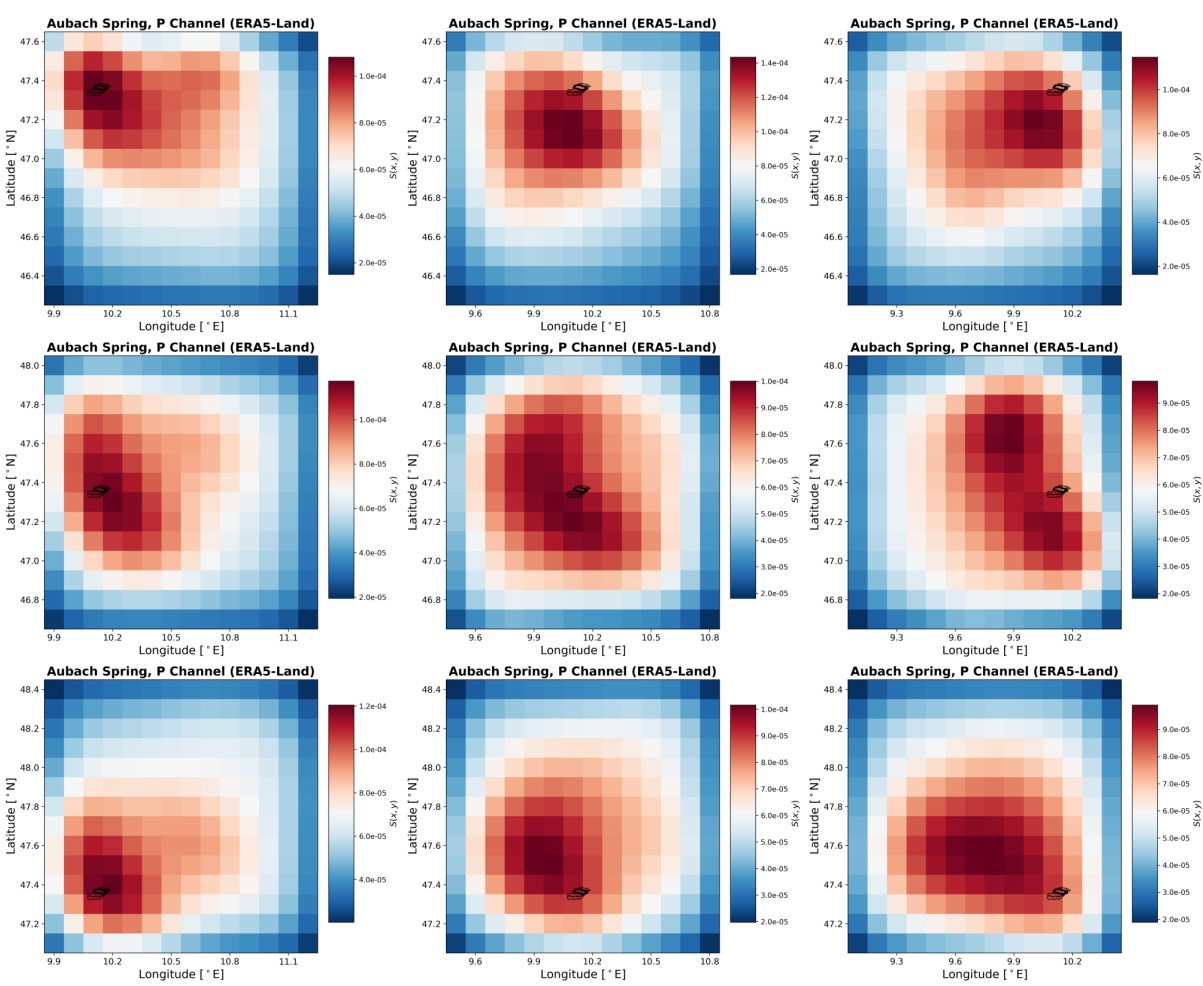

**Figure C2.** P-channel heatmaps based on ERA5-Land gridded data for Aubach spring with shifted area of the spatial input data in relation to the catchment position.

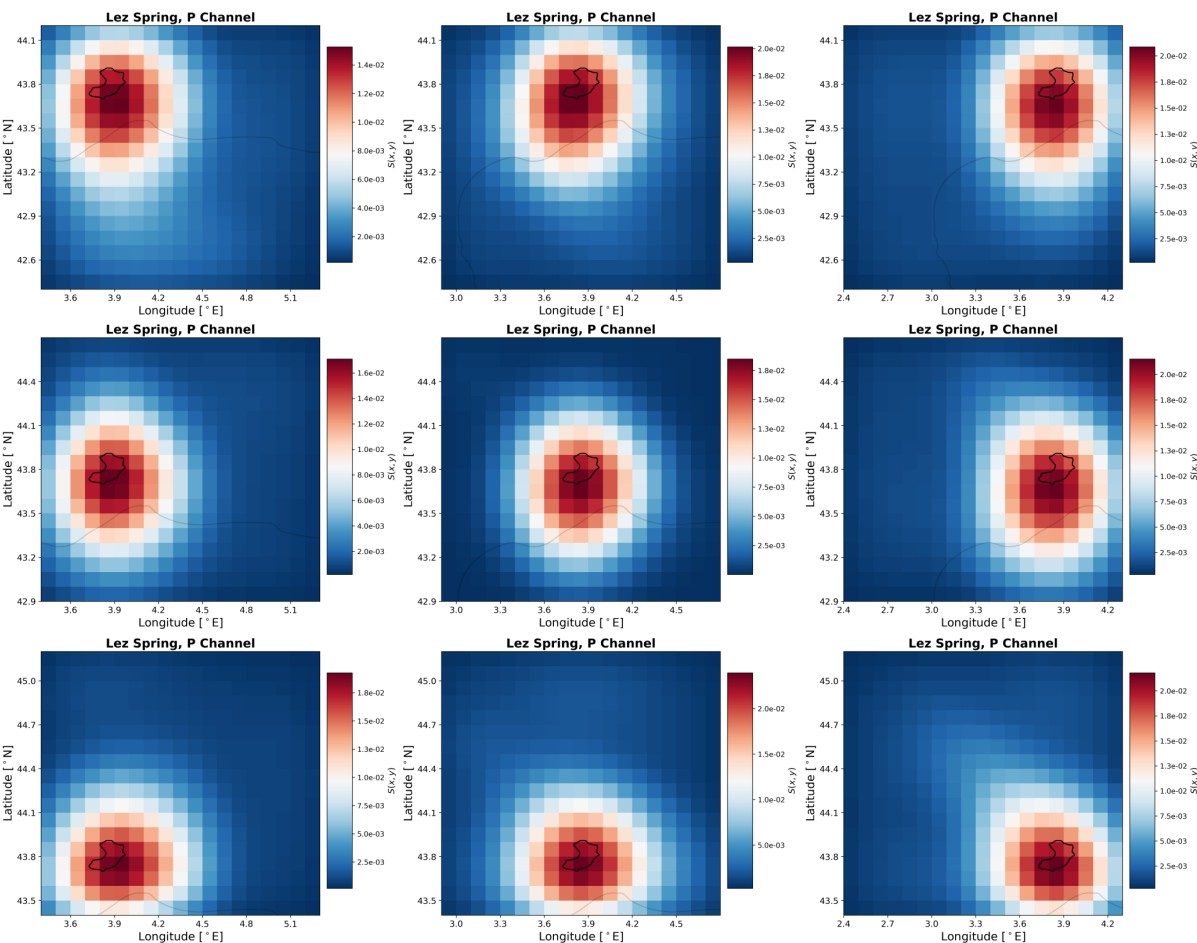

**Figure C3.** P-channel heatmaps based on E-OBS gridded data for Lez spring with shifted area of the spatial input data in relation to the catchment position.

## Appendix D: Model Overview

**Table D1.** Model parameter summary table.

| Optimized Hyperparamaters | Aubach (ERA5) | Aubach (RADOLAN) | Lez | Unica |
|---|---|---|---|---|
| n (1DConv filter) | 128 | 128 | 16 | 16 |
| input seq. length | 54 (hours) | 162 (hours) | 53 (days) | 40 (days) |
| batch-size | 64 | 256 | 32 | 32 |
| **Other Hyperparameters** | | | | |
| inital learning rate | 0.001 | 0.001 | 0.001 | 0.001 |
| training epochs | 100 | 100 | 100 | 100 |
| early stopping patience | 12 | 8 | 12 | 12 |
| **Model Summaries** | | | | |
| total parameters | 708,353 | 1,502,849 | 358,977 | 384,017 |
| trainable par. | 708,097 | 1,502,593 | 358,945 | 383,985 |
| non-trainable par. | 256 | 256 | 32 | 32 |

*Author contributions.* AW, TL and NG conceptualized the study, AW and TL developed the methodology and software code, and validated the results. AW performed the experiments, and investigated and visualized the results. GC and ZC performed formal analysis, NR and GC contributed to data curation activities. AW wrote the original paper draft with contributions from GC and NR. All authors contributed to interpretation of the results, and review and editing of the paper draft. TL and NG supervised the work.

*Competing interests.* The authors declare that they have no conflict of interest.

*Acknowledgements.* The financial support of KIT through the German Federal Ministry of Education and Research (BMBF) and the European Commission through the Partnership for Research and Innovation in the Mediterranean Area (PRIMA) program under Horizon 2020 (KARMA project, grant agreement number 01DH19022A) is gratefully acknowledged. We thank the French Ministry of Higher Education and Research for the thesis scholarship of G. Cinkus as well as the European Commission and the Agence Nationale de la Recherche (ANR) for its support of HSM and UMR through the Partnership for Research and Innovation in the Mediterranean Area (PRIMA) program under Horizon 2020 (KARMA project, ANR-18-PRIM-0005). We further acknowledge financial support by the Slovenian Research Agency within the project Infiltration processes in forested karst aquifers under changing environment (No. J2-1743). The authors acknowledge support by the state of Baden-Württemberg through bwHPC. Muñoz Sabater, J., (2019) was downloaded from the Copernicus Climate Change Service (C3S) Climate Data Store. The results contain modified Copernicus Climate Change Service information 2020. Neither the European Commission nor ECMWF is responsible for any use that may be made of the Copernicus information or data it contains. We acknowledge the E-OBS dataset and the data providers in the ECA&D project (https://www.ecad.eu), data from MeteoFrance, DWD and the office of the

federal state of Vorarlberg, division of water management. Lez spring discharge data were provided by the KARST observatory network (SNO KARST) initiative from the INSU/CNRS (FRANCE), which aims to strengthen knowledge-sharing and to promote cross-disciplinary research on karst systems. The authors further acknowledge support by the KIT-Publication Fund of the Karlsruhe Institute of Technology.

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
