# Peer review of "Karst spring discharge modeling based on deep learning using spatially distributed input data"

_Hydrology and Earth System Sciences, 2021_

## Author Comment (AC1)

**Response to Referee #1: by Kuo-Chin Hsu**

We thank Kuo-Chin Hsu for reviewing our manuscript and for the comments provided. We will address all aspects in the following. Please find our answers in red and the original comments in black.

The manuscript proposed to use convolutional neural network (CNN) associated with gridded meteorological data for Karst spring discharge modeling. CNN was applied to three karst spring watersheds in Europe. Results of 2D CNN model associated with gridded meteorological cells were compared to that of 1D CNN using climate station input data.
The manuscript is well written and technical sound.

General comments:

- CNN is a mature data-driven tool which highly relies on data availability and quality. The authors argue that less data is needed in the proposed approach to obtain satisficing results compared to previous deep learning approach and overcome the short of data from climate stations. The results show that 2D modeling is not necessary better than that of 1D and previous modeling in Lez spring. A question raised is that whether the key input data has been identified. For example, pumping may play an important driving factor but is not included in training and screened out by Bayesian model. Gridded meteorological data may not be enough to improve the model performance. The authors needs to address their contribution. Guide line for data preparation will be helpful for the suggestion of use machine learning.

Thank you for this thoughtful comment on data aspects. We think indeed that the 2D approach can overcome difficulties in climate station data availability. As we state in the manuscript: "climate stations are often not available within the catchment of a spring, do not match the data availability of the discharge time series (period or temporal resolution), or are more distant and thus do not truly represent the events in the catchment itself". Nevertheless, we want to clarify that we do not think that the 2D approach needs less data, instead we think that rather the amount of work necessary to collect and preprocess the data is strongly reduced. Gridded meteorological data is available online and needs only minor preprocessing in contrary to most climate station data.

We agree that the results of the 2D approach are not necessarily better, as it can be seen in the example of Lez spring that you mention. It is true that Lez spring discharge is a complex combination of natural discharge, pumping and legally regulated minimum discharge from the extracted water to protect downstream ecosystems. Nevertheless, we showed, that we were able to simulate the discharge with solely meteorological data, using both the 1D and 2D models. In this specific case of Lez spring, a lot of work was necessary to produce the 1D precipitation input time series (compare Appendix B - Lez Catchment Precipitation Interpolation) due to very patchy climate station data. It seems that these additional efforts pay off in terms of higher performance compared to the 2D model. Nevertheless, the 2D climate data seems to offer a sufficient substitution, if needed.

Regarding the consideration of other external driving factors like for example pumping, we agree that these may play an important role, and their inclusion might improve model performance. It would also be possible to combine the 2D input of meteorological parameters with a 1D-input of

additional parameters. However, the focus of our study was on the direct comparison of 1D and 2D-meteorological inputs, so we did not consider other than meteorological factors.

We do not think that we can provide a general guideline for data preparation, because this step strongly depends on the datasets that a specific user intends to use. Nevertheless, we have included a description of the input data format that is used in our published python scripts, which should enable future users to adapt and apply them.

- The modeling uncertainty is quite low to almost without uncertainty that seems abnormal. The authors may explain this.

We apologize that, given our current formulation, this aspect does not become clear. The shown model uncertainty is derived from an ensemble of 10 differently initialized models, each using Monte-Carlo dropout to produce an ensemble of 100 different forecasts (so 1000 in total). For each of the 10 models we calculated the 95% confidence interval of the 100 available forecasts (1.96 times standard deviation, because of sigma rule for 95% confidence). What is shown in the final plots, is the 95% uncertainty of the mean of all 10 model ensembles, which is indeed very small. In a revised version of the manuscript, we will shortly clarify this aspect. We want to add that the shown uncertainty does not include other sources of uncertainty of the models (such as input data uncertainty). For this reason, it might seem abnormally small to you at first glance. We also will add a clarifying statement on this aspect in a revised version of the manuscript.

Specific comments:

- Line 203, write the long short term memory for abbreviation of

Sorry, we forgot to introduce the abbreviation LSTM. We will correct that.

- Lines 213-215. Although the all programs are available from Python community. Technical functions should be described for the used library or framework should be explained.

We are not quite sure what you mean exactly. We tried to explain the most important technical functions (like CNN) in the methodology section and give the associated references. Given the limited space for the manuscript, we avoided an in-depth explanation of all details and hope that referencing the used packages and frameworks in combination with the published Python code is sufficient. Additional necessary information should become clear to the python-experienced user from our published code.

- It is not clear functions for training, validation, optimization and testing periods in Table 1. The should be explained in main text.

Yes, you are right. Sorry that we missed to explain the purpose of those sets. We will add a clarifying statement to a revised version of the manuscript in accordance with the comment on that aspect of Reviewer#2.

---

## Author Comment (AC2)

**Response to Anonymous Referee #2**

We thank the anonymous referee for a very comprehensive review of our manuscript and many very useful comments. A lot of important aspects were mentioned that we will use to improve our manuscript. We will address the specific aspects in the following. Please find our answers in red and the original comments in black.

This study uses convolutional neural networks (both 2D and 1D) to model streamflow in three karst spring catchments. They compare 1D CNNs, which process time series of weather station forcing data, to sequential 2D – 1D CNNs, which process gridded meteorological forcing data from climate reanalysis. The use of widely available gridded meteorological data is advantageous due to its more complete spatiotemporal availability, in comparison to data from weather stations.

This is an interesting study with potentially applicable results; however, there are several key limitations with its current form. The literature review at present is insufficient and does not provide enough information to explain the relevance or context of the results. In many cases, the likelihood of hypotheses and claims is stated without justification. Additionally, conclusions surrounding the potential of these models for karst catchment localization and delineation are currently overstepping the results shown.

Major comments

Overall, the introduction is very brief and does not provide adequate context for the current work. Johannet et al (1994) is referred to but it is not stated what they did. The authors can expand to consider the further history of ANNs in water related research (e.g. Hsu et al 1995, Maier and Dancy 1996, Zealand et al 1999, Maier and Dandy 2000 and the references therein), which can lead to the use of deeper networks in water research. Additionally, 1D CNNs have been used for streamflow prediction in the past by others who the authors do not refer to (e.g. Hussain et al 2020, Van et al 2020). By further fleshing out the relevant history the authors can more convincingly present the relevance and potential applications of their work.

We admit that the introduction is very brief, since we specifically focused on the application of CNN on karst spring discharge modeling. We do not think that a comprehensive introduction of ANN applications in water resources related research in general is possible nor necessary. However, in a revised version, we will provide more context of specific ANN and CNN applications in the areas of groundwater modeling and streamflow forecasting and as well on a broader water resources application context.

Why do the authors use a 1D CNN to learn temporal features rather than an LSTM, despite the many successes of LSTM-based modelling for streamflow prediction (e.g. Kratzert et al. 2018, 2019a, 2019b; Gauch et al. 2021, Frame et al. 2021, Anderson and Radic 2021; note that this list is not comprehensive or that all required but these papers can be a starting point for the authors to consider)? While there is a brief mention that 1D-CNNs are fast/stable, there is little mention or consideration given to the vast success of the LSTM approach at both daily and hourly scales, which is surprising given their prevalence. This study uses only three catchments and so I can't imagine that the training time between a 2D – 1D CNN model vs a CNN-LSTM model would be prohibitively different. There is

opportunity here to compare the two approaches quantitatively to see which performs better (e.g. perhaps a CNN-LSTM model will be able to simulate the streamflow peaks around Oct 2020 in Figure 3), or if there are differences in the "catchment delineation" results.  While the authors don't absolutely need to perform this comparison, it would certainly help to justify their methodological choices (2D-1D CNN vs CNN-LSTM).

Thank you for this comment, which is completely understandable, especially given the successes of LSTM models, some of which you mentioned yourself. We will give an extended justification of model choices for the 1D-CNN models that we used in a revised version of our manuscript. In preliminary work we did indeed also test LSTM models in combination with 2DCNNs and you are correct, for the 2D-models there is no large difference in training time (at least for these three sites) between 2DCNN+1DCNN and 2DCNN+LSTM. However, we also did not find systematic superiority or performance differences between LSTM and 1DCNNs as subsequent models to the 2DCNNs. Moreover, we also have shown in the closely related application of groundwater level forecasting, that 1D CNNs have in most cases an equal performance to LSTMs (Wunsch et al., 2021). Van et al (2020) also show the potential of CNNs compared to LSTMs in case of rainfall runoff modeling.

To be methodologically more consistent, we therefore decided to neglect 2DCNN+LSTM models. This way, we do not change the forecasting model itself, but only replace the climate station input data by a 2D-CNN model, which learns relevant data itself. The primary goal of our study was to show that spatially distributed climate data can act as a reliable substitute in case of bad availability of climate station data. By not mixing up different model types, in our opinion a comparison between 1D and 2D-based models is more convincing.  Though it would of course be possible to compare the CNN vs. the LSTM approach quantitatively to see which performs better, we would like to avoid this, as it would be just a different kind of research aspect and we prefer to keep the focus and the comparison of 1D vs. 2D meteorological inputs.

References:

Van, S. P., Le, H. M., Thanh, D. V., Dang, T. D., Loc, H. H., and Anh, D. T.: Deep learning convolutional neural network in rainfall–runoff modelling, Journal of Hydroinformatics, 22, 541–561, https://doi.org/10/ggskkh, 2020.

Wunsch, A., Liesch, T., and Broda, S.: Groundwater level forecasting with artificial neural networks: a comparison of long short-term memory (LSTM), convolutional neural networks (CNNs), and non-linear autoregressive networks with exogenous input (NARX), 25, 1671–1687, https://doi.org/10.5194/hess-25-1671-2021, 2021.

In many cases, the authors assert the likelihood of a claim without justification.  These instances either need further exploration or to be reframed as "possible" rather than "probable".  For example, in line 297 it is stated that the main source of uncertainty is "probably the uncertainty of parameter values resulting from the ERA5-land grid cell sizes".  How are the authors confident that this is "probable"? Are the authors referring to the uncertainty, or error?  There are other places where the authors describe a result or hypothesis as being "probable" without any justification.  These instances are more conjecture than a discussion of probability (e.g. line 344, line 359, line 363, among others) and are not very convincing without additional support.  Furthermore, there are instances where highly certain language is used without quantification (e.g. "perfectly" in line 297) or a contradictory mixture of

language is used (e.g. "probably perfectly" in line 360; "quite exactly" in line 411). These should be changed as well. Another example is in line 407, where the authors suggest that the shape of the sensitivity pattern may be a "relic from spatial correlation of precipitation events". Can this hypothesis be supported or explored with evidence? The authors have the needed precipitation data so I am not sure why this conjecture is here without support.

Thank you for this important comment. We apologize for our inaccurate wording. We will revise our manuscript accordingly and will provide justification where necessary and possible.

I have concerns that the sensitivity methods applied do not actually work to "delineate" or "identify" the catchments to the degree that the authors are claiming. One key difference between Anderson and Radic (2021) and this study is that the basins in Anderson and Radic (2021) are in different regions of the input space, while the basins here are all centered (Figure 6). Having stations in different regions of the input was key to interpret if the model was learning to focus on the right regions in Anderson and Radic (2021); e.g. in Figure 7 in Anderson and Radic (2021), the model is sensitive in different regions which tells us that the model is learning different things for different stations. In Figures 6 and C1, one could argue that the models here have learned to generally focus on the central area under all circumstances and it is just coincidence that the basins are centered there as well. As I see it, in order to make steps towards catchment delineation, the authors need to demonstrate that the model automatically focuses on (1) the right location and (2) have the right "area of sensitivity". Neither of these points are quantified in this work, while both are referred to qualitatively; however, both can (and should) be more rigorously investigated. To point (1), the authors could run different tests with the basins placed at different locations in the input (e.g. 9 (or more) models could be made for each basin – one with the basin located in the top left area of the input, one with the basin located in the top center area of the input, etc). Then, the location of maximum sensitivity can be quantified and compared to the location of the basin. In this way (or in some similar way), the authors could more concretely conclude whether their approach is learning to focus on the correct area of the input or not. To point (2), the authors could run different tests with different input areas (e.g. for each basin, the authors could double/triple/etc the number of pixels in the x- and y- directions). Then, the "most sensitive area" can be quantified (e.g. the area greater than the half-maximum sensitivity, or some alternative metric) and compared with the basin area. The authors can find: is the most sensitive area always comparable to the known basin area? By addressing these two points (either as described above or in some other way), the authors can more convincingly state whether the CNN approach has potential for catchment delineation.

We understand your concerns. We have already started to perform additional model runs to show that our models can learn the correct location, regardless of the position of the catchment. Please find a preliminary result for Unica springs in the following:

[Figure]

We will add the results of this analyses to the revised manuscript.

Regarding point (2): Currently we think that this point is hard to evaluate. A threshold of the "most sensitive area" would hardly be transferable a) between parameters (as it strongly depends on the autocorrelation of different inputs, e.g. T is usually more auto-correlated than P) and b) between specific regions (in mountainous regions, meteorological parameters are less auto-correlated than in flat landscapes). Though this is an interesting idea, and we will keep in mind for future research, we do not think it is possible to include it in the current study.

Additional comments:

Paragraph starting at line 43: This paragraph can be hard to follow when very little has been done to describe the architectures (e.g. the acronym 'LSTM' has not been defined).
Thank you, we will better introduce LSTMs in a revised version of the manuscript.

Line 55: The authors state ANNs to be superior for points i) and iii), but no justification is given as to why they expect this.
Indeed, our formulation is not well justified and therefore misleading. In fact, we think that the most important advantage of our ANN-approach over a pure event-correlation is that ANNs are able to represent non-linear relationships. We will reformulate this part in the revised manuscript.

Line 200: Add references for batch normalization (e.g. Loffe et al, 2015) and dropout (e.g. Srivastava et al, 2014)
Thank you. Will be done.

Line 203: LSTMs are claimed to be slower and with similar performance as compared to 1D CNNs for "this specific application". Does that mean that the authors have used LSTMs for streamflow modelling in karst catchments as well?
As mentioned above, we have compared LSTMs and CNNs in preliminary work of this study (i.e. for karst spring modelling) and we also compared them for the closely related application of groundwater level forecasting. We will clarify this in the revised manuscript.

Section 3.2: It would be very useful to have an overview of the models that were used. Currently in Table 1 there is the time splitting scheme. In addition, it should be more clearly listed the length of the input sequence, number of observations, and number of parameters in each model (or layer).
Thank you for this comment. This information is indeed missing. We will provide it in a revised version or at least in a Supplement.

Section 3.4: This section is very brief and does provide much context for the methods chosen (e.g. why follow Anderson and Radic, and not other interpretability methods?). Some statements are vague (e.g. "In short it works by perturbing spatial fractions of the input data by using a 2D-Gaussian curve" – what is meant by 'using'?). The final few sentences are written with certainty, although the methods have not been applied yet (e.g. "... a smaller area will most certainly have a higher influence on the spring discharge..."). This section can be challenging to follow, and I suspect especially so for readers who are not as familiar with Anderson and Radic (2021).

We are sorry that we obviously do not provide sufficient context. We will modify this section to better clarify what was done by Anderson & Radic and to what we refer. Regarding the visualization of input importance, we follow the approach of Anderson and Radic, because we think it is logical and yet simple, and it proved to be appropriate in a similar context. The method chosen, should be able to handle time series aspects, such as different areas being important for different time steps. Other interpretability methods are mostly applied to classification problems. Therefore, perturbing the input, instead of using activation and gradient methods, or saliency maps (gradient ascent), seems to be a valid approach to capture the time variance of the input importance.

Concerning the final few sentences "… a smaller area will most certainly have a higher influence on the spring discharge…": We conclude that from the aspect of different spatial auto-correlation of the input parameters (what we meant with "very different spatial heterogeneity and variability") which is usually higher for T than for P – thus it seems logic that the "sensitive area" is larger for T and smaller for P. Overall, this was meant as a justification for our modification of the original method. We will try to better explain this in a revised version.

Line 254: How is "satisfying fit" qualified?  Satisfying as compared to what?

We meant compared to the usual range of NSE and $R^2$ for different modelling approaches for Karst springs and water resources modeling in general, where an NSE > 0.65 is regarded as satisfying. We will be more precise in our wording in a revised version of our manuscript.

Line 276: It is not surprising that the models are within the same "order of magnitude".
An order of magnitude of NSE spans 0.1 through 1, which is a huge range of performance.

We apologize for that mistake, which comes from inadequate translation of a figure of speech. We meant "in the same range". We will correct this.

Sections 4.1 – 4.3: These sections are a mixture of results and discussion.  While that is not inherently an issue, both results and discussion are mixed throughout each section in ways that vary from section to section (e.g. 4.1 begins with results, but 4.2 begins with discussion before even a description of Figure 4).  It would be easier to read and follow if the authors were more consistent between sections (e.g. first have results of 4.1, then discussion of 4.1, then results of 4.2 etc).

Thank you for this hint. We will restructure the Results and Discussion part accordingly.

Line 331: It seems the authors mean "substantially" and not "significantly" as there is no discussion of statistical significance here.  "Significant" is also referred to in author places where the authors do not appear to mean it in the formal sense.

We apologize for unprecise wording and will adapt our manuscript accordingly in a revised version.

Line 339: If this new model is not going to be discussed or explored clearly, then it should not be brought up at all.

Thank you for this comment. We will remove this part from a revised version.

Lines 371 – 377: This is description of other studies should be moved to the introduction.

We will reconsider both the structures of our discussion and the according part in the introduction. In its current form we do not introduce the study areas in the introduction, but we will think of an appropriate placement of these studies.

Section 4.4 (and Section 3.4): It is not clear how the authors are defining catchment delineation/identification. Are they referring to the areas of the sensitivity fields that are greater than the half-maximum of sensitivity? For which input variable? It should be clarified just how the step from sensitivity heat map to catchment delineation could be made.

Currently we do not quantitively evaluate the sensitivity heatmaps, but judge only visually, if the known catchment coincides well with the high-sensitive areas on the heatmaps. So maybe "identifying the approximate catchment location" is a better formulation than "catchment delineation" in the present state. We will clarify this in a revised version. Primary purpose of this study was to show the usefulness of the 2D approach for discharge forecasting. Identifying the catchment location was a very interesting additional aspect, which seems to be worth investigating further in the future. We will add a short discussion of possibilities for catchment delineation that should be explored in future studies.

References

Frame, J., Kratzert, F., Klotz, D., Gauch, M., Shelev, G., Gilon, O., Qualls, L. M., Gupta, H. V., and Nearing, G. S.: Deep learning rainfall-runoff predictions of extreme events, Hydrol. Earth Syst. Sci. Discuss. [preprint], https://doi.org/10.5194/hess-2021-423, in review, 2021.

Gauch, M., Kratzert, F., Klotz, D., Nearing, G., Lin, J., and Hochreiter, S.: Rainfall–runoff prediction at multiple timescales with a single Long Short-Term Memory network. Hydrol. Earth Syst. Sci. https://doi.org/10.5194/hess-25-2045-2021, 2021.

Hsu, K., Gupta, H. V. and Sorooshian, S.: Artificial Neural Network Modeling of the RainfallRunoff Process, Water Resour. Res., 31(10), 2517–2530, doi:https://doi.org/10.1029/95WR01955, 1995.

Hussain, D., Hussain, T., Khan, A. A., Naqvi, S. A. A. and Jamil, A.: A deep learning approach for hydrological time-series prediction: A case study of Gilgit river basin, Earth Sci. Informatics, 13(3), 915–927, doi:10.1007/s12145-020-00477-2, 2020.

Kratzert, F., Klotz, D., Brenner, C., Schulz, K. and Herrnegger, M.: Rainfall–runoff modelling using Long Short-Term Memory (LSTM) networks, Hydrol. Earth Syst. Sci., 22(11), 6005–6022, doi:10.5194/hess-22-6005-2018, 2018.

Kratzert, F., Klotz, D., Herrnegger, M., Sampson, A. K., Hochreiter, S. and Nearing, G. S.: Toward Improved Predictions in Ungauged Basins: Exploiting the Power of Machine Learning, Water Resour. Res., 55(12), 11344–11354, doi:10.1029/2019WR026065, 2019a.

Kratzert, F., Klotz, D., Shalev, G., Klambauer, G., Hochreiter, S. and Nearing, G.: Towards learning universal, regional, and local hydrological behaviors via machine learning applied to large-sample datasets, Hydrol. Earth Syst. Sci., 23(12), 5089–5110, doi:http://dx.doi.org/10.5194/hess-23-5089-2019, 2019b.

Loffe, S. and Szegedy, C.: Batch Normalization: Accelerating Deep Network Training by Reducing Internal Covariate Shift. arXiv, 2015.

Maier, H. R. and Dandy, G. C.: The Use of Artificial Neural Networks for the Prediction of Water Quality Parameters, Water Resour. Res., 32(4), 1013–1022, doi:10.1029/96WR03529, 1996.

Maier, H. R. and Dandy, G. C.: Neural networks for the prediction and forecasting of water resources variables: a review of modelling issues and applications, Environ. Model. Softw., 15(1), 101–124, doi:https://doi.org/10.1016/S1364-8152(99)00007-9, 2000.

Srivastava, N., Hinton, G., Krizhevsky, A., Sutskever, I. and Salakhutdinov, R.: Dropout: A Simple Way to Prevent Neural Networks from Overfitting, J. Mach. Learn. Res., 15(1), 1929–1958, 2014.

Van, S. P., Le, H. M., Thanh, D. V., Dang, T. D., Loc, H. H. and Anh, D. T.: Deep learning convolutional neural network in rainfall–runoff modelling, J. Hydroinformatics, 22(3), 541–561, doi:10.2166/hydro.2020.095, 2020.

Zealand, C. M., Burn, D. H. and Simonovic, S. P.: Short term streamflow forecasting using artificial neural networks, J. Hydrol., 214(1), 32–48, doi:https://doi.org/10.1016/S0022-1694(98)00242-X, 1999.

---

## Author Response (AR1)

We thank all Referees for reviewing our manuscript. We appreciate the constructive comments and will provide a point by point answer in the following. Please find the original comments in black and our answers in red. Line Numbers in our answers refer to the revised version of the manuscript.

**Comments by the Editor:**

Dear authors,

The two reviewers have completed the reviewing and pointed out several key limitations of this study. After carefully reading the manuscript, unfortunately, I have the similar opinions. Therefore, I expect that the authors take these comments seriously when they revise the manuscript, especially the follow comments:

1) A substantial revision of the introduction is needed by expanding to consider the further history of ANNs in water related research.
2) The choice of ANN model structure needs to be further justified.
3) The authors need to be careful about their claims without justification.
4) As you stated in your response, Lez spring discharge is a complex combination of natural discharge, pumping and legally regulated minimum discharge from the extracted water to protect downstream ecosystem. Therefore, the identification of the key input data for the ANN models is substantial.

We thank the Editor for this assessment. We do not answer to the points in detail, because they refer to referee comments, which are discussed and answered in detail in the following.

**Response to Referee #1: by Kuo-Chin Hsu**

The manuscript proposed to use convolutional neural network (CNN) associated with gridded meteorological data for Karst spring discharge modeling. CNN was applied to three karst spring watersheds in Europe. Results of 2D CNN model associated with gridded meteorological cells were compared to that of 1D CNN using climate station input data.

The manuscript is well written and technical sound.

General comments:

- CNN is a mature data-driven tool which highly relies on data availability and quality. The authors argue that less data is needed in the proposed approach to obtain satisficing results compared to previous deep learning approach and overcome the short of data from climate stations. The results show that 2D modeling is not necessary better than that of 1D and previous modeling in Lez spring. A question raised is that whether the key input data has been identified. For example, pumping may play an important driving factor but is not included in training and screened out by Bayesian model. Gridded meteorological data may not be enough to improve the model performance. The authors needs to address their contribution. Guide line for data preparation will be helpful for the suggestion of use machine learning.

Thank you for this thoughtful comment on data aspects. We think indeed that the 2D approach can overcome difficulties in climate station data availability. As we state in the manuscript: "climate stations are often not available within the catchment of a spring, do not match the data availability of the discharge time series (period or temporal resolution), or are more distant and thus do not truly represent the events in the catchment itself". Nevertheless, we want to clarify that we do not think that the 2D approach needs less data, instead we think that rather the amount of work necessary to collect and preprocess the data is strongly reduced. Gridded meteorological data is available online and needs only minor preprocessing in contrary to most climate station data.

We agree that the results of the 2D approach are not necessarily better, as it can be seen in the example of Lez spring that you mention. It is true that Lez spring discharge is a complex combination of natural discharge, pumping and legally regulated minimum discharge from the extracted water to protect downstream ecosystems. Pumping is performed directly in the karst aquifer, thus also during dry periods with zero discharge, the legally regulated minimum discharge however is part of this extracted water. The legally regulated minimum discharge is released downstream into the Lez river (after the spring) and basically has no effect on the spring discharge other than being part of the pumped water volume. Please see the following illustration:

[Figure]

Despite all these factors, we showed, that pumping is not a necessary input, to achieve highly accurate (NSE and KGE > 0.86 for the 1D model and > 0.75 for the 2D model) results for Lez spring and that we were able to simulate the discharge with solely meteorological data, using both the 1D and 2D models. We therefore think that focusing on climatic inputs allows a more consistent approach for all three study areas. Moreover, the focus of our study was to compare 1D meteorological inputs of climate stations to gridded 2D meteorological inputs, and not to achieve the best performance for each of the test sites. Another advantage of not

including pumping data is that our approach can potentially be transferred on real forecasting tasks using weather/climate forecasts on different time scales in the future. For such applications, future pumping rates are not available. The most important reason not to include pumping is that the available data begins in 2011, which means we would loose three years of data in total for an already comparably short time series. We have now added a detailed explanation of not considering pumping as an input parameter to the manuscript. (L 160f, 407ff.)

In this specific case of Lez spring, a lot of work was necessary to produce the 1D precipitation input time series (compare Appendix B - Lez Catchment Precipitation Interpolation) due to very patchy climate station data. It seems that these additional efforts pay off in terms of higher performance compared to the 2D model. Nevertheless, the 2D climate data seems to offer a sufficient substitution, if needed.

We do not think that we can provide a general guideline for data preparation, because this step strongly depends on the datasets that a specific user intends to use. Nevertheless, we have included a description of the input data format that is used in our published python scripts, which should enable future users to adapt and apply them.

- The modeling uncertainty is quite low to almost without uncertainty that seems abnormal. The authors may explain this.

We apologize that, given our current formulation, this aspect does not become clear. The shown model uncertainty is derived from an ensemble of 10 differently initialized models, each using Monte-Carlo dropout to produce an ensemble of 100 different forecasts (so 1000 in total). For each of the 10 models we calculated the 95% confidence interval of the 100 available forecasts (1.96 times standard deviation, because of sigma rule for 95% confidence). What is shown in the final plots, is the 95% uncertainty of the mean of all 10 model ensembles, which is indeed very small. In the revised version of the manuscript, we have shortly clarified this aspect (Line 231ff.). We want to add that the shown uncertainty does not include other sources of uncertainty of the models (such as input data uncertainty). For this reason, it might seem abnormally small to you at first glance. We have also added a clarifying statement on this aspect in the revised version of the manuscript (Lines 300-302).

Specific comments:

- Line 203, write the long short term memory for abbreviation of

The LSTM as abbreviation is now introduced in the introduction. Thank you for pointing out. (L. 48-50)

- Lines 213-215. Although the all programs are available from Python community. Technical functions should be described for the used library or framework should be explained.

We are not quite sure what you mean exactly. We tried to explain the most important technical functions (like CNN) in the methodology section and give the associated references. Given the limited space for the manuscript, we avoided an in-depth explanation of all details and hope that referencing the used packages and frameworks in combination with the published Python code is sufficient. Additional necessary information should become clear to the python-experienced user from our published code.

- It is not clear functions for training, validation, optimization and testing periods in Table 1. The should be explained in main text.

Yes, you are right. Sorry that we missed to explain the purpose of those sets. We added a clarifying statement in Lines 242-245 in accordance with the comment on that aspect of Reviewer#2.

**Response to Anonymous Referee #2**

This study uses convolutional neural networks (both 2D and 1D) to model streamflow in three karst spring catchments. They compare 1D CNNs, which process time series of weather station forcing data, to sequential 2D – 1D CNNs, which process gridded meteorological forcing data from climate reanalysis. The use of widely available gridded meteorological data is advantageous due to its more complete spatiotemporal availability, in comparison to data from weather stations.

This is an interesting study with potentially applicable results; however, there are several key limitations with its current form. The literature review at present is insufficient and does not provide enough information to explain the relevance or context of the results. In many cases, the likelihood of hypotheses and claims is stated without justification. Additionally, conclusions surrounding the potential of these models for karst catchment localization and delineation are currently overstepping the results shown.

Major comments

Overall, the introduction is very brief and does not provide adequate context for the current work. Johannet et al (1994) is referred to but it is not stated what they did. The authors can expand to consider the further history of ANNs in water related research (e.g. Hsu et al 1995, Maier and Dancy 1996, Zealand et al 1999, Maier and Dandy 2000 and the references therein), which can lead to the use of deeper networks in water research. Additionally, 1D CNNs have been used for streamflow prediction in the past by others who the authors do not refer to (e.g. Hussain et al 2020, Van et al 2020). By further fleshing out the relevant history the authors can more convincingly present the relevance and potential applications of their work. We admit that the introduction was very brief, since we specifically focused on the application of CNN on karst spring discharge modeling. We now have added a statement on the study of Johannet et al. (1994) and extended the literature review on ANN application in water resources related research. We hope that this now better fulfills the requirements of presenting the relevance of our work more convincingly. Further we now provide references of 1D-CNNs in groundwater and streamflow/runoff modeling (Lines 22f., L25ff).

Why do the authors use a 1D CNN to learn temporal features rather than an LSTM, despite the many successes of LSTM-based modelling for streamflow prediction (e.g. Kratzert et al. 2018, 2019a, 2019b; Gauch et al. 2021, Frame et al. 2021, Anderson and Radic 2021; note that this list is not comprehensive or that all required but these papers can be a starting point for the authors to consider)? While there is a brief mention that 1D-CNNs are fast/stable, there is little mention or consideration given to the vast success of the LSTM approach at both daily and hourly scales, which is surprising given their prevalence. This study uses only three catchments and so I can't imagine that the training time between a 2D – 1D CNN model vs a

CNN-LSTM model would be prohibitively different. There is opportunity here to compare the two approaches quantitatively to see which performs better (e.g. perhaps a CNN-LSTM model will be able to simulate the streamflow peaks around Oct 2020 in Figure 3), or if there are differences in the "catchment delineation" results. While the authors don't absolutely need to perform this comparison, it would certainly help to justify their methodological choices (2D-1D CNN vs CNN-LSTM).

Thank you for this comment, which is completely understandable, especially given the successes of LSTM models, some of which you mentioned yourself. In preliminary work we did indeed also test LSTM models in combination with 2DCNNs and you are correct, for the 2D-models there is no large difference in training time (at least for these three sites) between 2DCNN+1DCNN and 2DCNN+LSTM. However, we also did not find systematic superiority or performance differences between LSTM and 1DCNNs as subsequent models to the 2DCNNs. Moreover, we also have shown in the closely related application of groundwater level forecasting, that 1D CNNs have in most cases an equal performance to LSTMs (Wunsch et al., 2021). Van et al (2020) also show the potential of CNNs compared to LSTMs in case of rainfall runoff modeling.

To be methodologically consistent, we therefore decided to neglect 2DCNN+LSTM models. This way, we do not change the forecasting model itself, but only replace the climate station input data by a 2D-CNN model, which learns relevant data itself. The primary goal of our study was to show that spatially distributed climate data can act as a reliable substitute in case of bad availability of climate station data. By not mixing up different model types, in our opinion a comparison between 1D and 2D-based models is more convincing. Though it would of course be possible to compare the CNN vs. the LSTM approach quantitatively to see which performs better, we would like to avoid this, as it would be just a different kind of research aspect and we prefer to keep the focus and the comparison of 1D vs. 2D meteorological inputs.

We have extended the justification of model choices in Lines 219-230.

References:

Van, S. P., Le, H. M., Thanh, D. V., Dang, T. D., Loc, H. H., and Anh, D. T.: Deep learning convolutional neural network in rainfall–runoff modelling, Journal of Hydroinformatics, 22, 541–561, https://doi.org/10/ggskkh, 2020.

Wunsch, A., Liesch, T., and Broda, S.: Groundwater level forecasting with artificial neural networks: a comparison of long short-term memory (LSTM), convolutional neural networks (CNNs), and non-linear autoregressive networks with exogenous input (NARX), 25, 1671–1687, https://doi.org/10.5194/hess-25-1671-2021, 2021.

In many cases, the authors assert the likelihood of a claim without justification. These instances either need further exploration or to be reframed as "possible" rather than

"probable".  For example, in line 297 it is stated that the main source of uncertainty is "probably the uncertainty of parameter values resulting from the ERA5-land grid cell sizes". How are the authors confident that this is "probable"?  Are the authors referring to the uncertainty, or error?  There are other places where the authors describe a result or hypothesis as being "probable" without any justification.  These instances are more conjecture than a discussion of probability (e.g. line 344, line 359, line 363, among others) and are not very convincing without additional support.  Furthermore, there are instances where highly certain language is used without quantification (e.g. "perfectly" in line 297) or a contradictory mixture of language is used (e.g. "probably perfectly" in line 360; "quite exactly" in line 411). These should be changed as well.  Another example is in line 407, where the authors suggest that the shape of the sensitivity pattern may be a "relic from spatial correlation of precipitation events".  Can this hypothesis be supported or explored with evidence?  The authors have the needed precipitation data so I am not sure why this conjecture is here without support.

Thank you for pointing out. We indeed used inaccurate wording at several passages, which we have revised throughout the complete manuscript. Please see the track changes document for a convenient overview of the changed wording.
See for example Lines: 290f., 300f., 309f., 372f., 375f., 391f., 395f., 421f., and others

Please also find our statement on the shape of the sensitivity pattern at Aubach spring in Lines 432-440.

I have concerns that the sensitivity methods applied do not actually work to "delineate" or "identify" the catchments to the degree that the authors are claiming.  One key difference between Anderson and Radic (2021) and this study is that the basins in Anderson and Radic (2021) are in different regions of the input space, while the basins here are all centered (Figure 6).  Having stations in different regions of the input was key to interpret if the model was learning to focus on the right regions in Anderson and Radic (2021); e.g. in Figure 7 in Anderson and Radic (2021), the model is sensitive in different regions which tells us that the model is learning different things for different stations.  In Figures 6 and C1, one could argue that the models here have learned to generally focus on the central area under all circumstances and it is just coincidence that the basins are centered there as well.  As I see it, in order to make steps towards catchment delineation, the authors need to demonstrate that the model automatically focuses on (1) the right location and (2) have the right "area of sensitivity". Neither of these points are quantified in this work, while both are referred to qualitatively; however, both can (and should) be more rigorously investigated.  To point (1), the authors could run different tests with the basins placed at different locations in the input (e.g. 9 (or more) models could be made for each basin – one with the basin located in the top left area of the input, one with the basin located in the top center area of the input, etc).  Then, the location of maximum sensitivity can be quantified and compared to the location of the basin. In this way (or in some similar way), the authors could more concretely conclude whether their approach is learning to focus on the correct area of the input or not.  To point (2), the

authors could run different tests with different input areas (e.g. for each basin, the authors could double/triple/etc the number of pixels in the x- and y- directions). Then, the "most sensitive area" can be quantified (e.g. the area greater than the half-maximum sensitivity, or some alternative metric) and compared with the basin area. The authors can find: is the most sensitive area always comparable to the known basin area? By addressing these two points (either as described above or in some other way), the authors can more convincingly state whether the CNN approach has potential for catchment delineation.

We understand your concerns that the model might just learn that the centering region is important. We followed your advice and performed additional analyses on all models, except the fine-resolution model for Aubach spring. This would have exceeded our computational possibilities for the short time available for revising this manuscript. We decided to focus exemplarily on the results of Unica spring in the manuscript, which nicely demonstrate that our models can indeed learn the correct location, regardless of the position of the catchment (Figure 7, Lines 458-478). For the sake of completeness, we added selected results for Lez spring and Aubach spring to the appendix (Figures C2 and C3). We now explain, what potential our approach bears for catchment delineation but also more clearly communicate that in its current form it is only useful for roughly localizing the position of the catchment (Lines 12, 67-68, title of section 3.4, 281-282, 455-456, Figure 7

Regarding point (2) of your comment: We think that this point is hard to evaluate given the results obtain in this study. A threshold (such as the half-maximum sensitivity you proposed) of the "most sensitive area" would hardly be transferable a) between parameters (as it strongly depends on the autocorrelation of different inputs, e.g. T is usually more auto-correlated than P) and b) between specific regions (in mountainous regions, meteorological parameters are less auto-correlated than in flat landscapes). Though this is an interesting idea, and we will keep in mind for future research, we do not think it is possible to include it in the current study. We rather think that a follow-on study with more catchments of adequate size should be performed, with a special focus on the delineation strategy development (Lines 487-490).

Additional comments:

Paragraph starting at line 43: This paragraph can be hard to follow when very little has been done to describe the architectures (e.g. the acronym 'LSTM' has not been defined).

Thank you, we now introduced the LSTM acronym in the introduction and rephrased this paragraph, to improve its understandability. L 47-57

Line 55: The authors state ANNs to be superior for points i) and iii), but no justification is given as to why they expect this.

Indeed, our formulation is not well justified and therefore misleading. In fact, we think that the most important advantage of our ANN-approach over a pure event-correlation is that ANNs are able to represent non-linear relationships. We rephrased this sentence. L 56-60

Line 200: Add references for batch normalization (e.g. Loffe et al, 2015) and dropout (e.g. Srivastava et al, 2014)

Thank you. We have added these references (L. 216, 217)

Line 203: LSTMs are claimed to be slower and with similar performance as compared to 1D CNNs for "this specific application". Does that mean that the authors have used LSTMs for streamflow modelling in karst catchments as well?

As mentioned above, regarding the extend justification of choice of methods, we have rephrased this paragraph. We hope it becomes clearer now, which experience and studies we draw our conclusions from. (L 219-228)

Section 3.2: It would be very useful to have an overview of the models that were used. Currently in Table 1 there is the time splitting scheme. In addition, it should be more clearly listed the length of the input sequence, number of observations, and number of parameters in each model (or layer).

Thank you for pointing out this important aspect. We added the number of observations to Table 1. Hyperparameters and information on the number of model parameters is now available in the appendix (Table D1).

Section 3.4: This section is very brief and does provide much context for the methods chosen (e.g. why follow Anderson and Radic, and not other interpretability methods?). Some statements are vague (e.g. "In short it works by perturbing spatial fractions of the input data by using a 2D-Gaussian curve" – what is meant by 'using'?). The final few sentences are written with certainty, although the methods have not been applied yet (e.g. "… a smaller area will most certainly have a higher influence on the spring discharge…"). This section can be challenging to follow, and I suspect especially so for readers who are not as familiar with Anderson and Radic (2021).

We apologize for not providing sufficient context. We have improved the section, added justification for following the approach of Anderson and Radic (2021), clarified the wording ("using") and relativized the certainty of our last statements. We also better explain why we modified their sensitivity approach to perturb only single input channels at a time. We hope the whole section is now easier to follow and more precise. L 268-282

Line 254: How is "satisfying fit" qualified?  Satisfying as compared to what?

Thank you for pointing out. We removed this inaccurate wording and rephrased the section. Furthermore, we revised the complete manuscript and replaced inaccurate wording.

L 286. We also changed e.g. L328ff.

Line 276: It is not surprising that the models are within the same "order of magnitude".

An order of magnitude of NSE spans 0.1 through 1, which is a huge range of performance.

We apologize for that mistake, which comes from inadequate translation of a figure of speech. We meant "in the same range". We corrected the wording accordingly. L 310

Sections 4.1 – 4.3: These sections are a mixture of results and discussion.  While that is not inherently an issue, both results and discussion are mixed throughout each section in ways that vary from section to section (e.g. 4.1 begins with results, but 4.2 begins with discussion before even a description of Figure 4).  It would be easier to read and follow if the authors were more consistent between sections (e.g. first have results of 4.1, then discussion of 4.1, then results of 4.2 etc).

Thank you for pointing out. We have restructured this part, to provide a consistent presentation of results and discussion. Please see Section 4.2 in the tracked-changes document.

Line 331: It seems the authors mean "substantially" and not "significantly" as there is no discussion of statistical significance here.  "Significant" is also referred to in author places where the authors do not appear to mean it in the formal sense.

Thank you for pointing out. We replaced several inaccurate usages of "significant" throughout the manuscript. Please see the tracked-changes document for all changes and for example the following lines: 122, 177, 335,  348, 354, 367, 385, 407, 494, 504

Line 339: If this new model is not going to be discussed or explored clearly, then it should not be brought up at all.

We agree, thank you. We removed this part from the revised version.

Lines 371 – 377: This is description of other studies should be moved to the introduction.

We admit that this is not the best place for describing the studies. We moved part of this description to the study area section, but also mention these studies now in the introduction (Lines 162ff., 62ff. ). We hope that you agree that it is now appropriate.

Section 4.4 (and Section 3.4): It is not clear how the authors are defining catchment delineation/identification. Are they referring to the areas of the sensitivity fields that are greater than the half-maximum of sensitivity?  For which input variable?  It should be clarified just how the step from sensitivity heat map to catchment delineation could be made.

Currently we do not quantitively evaluate the sensitivity heatmaps, but judge only visually, if the known catchment coincides well with the high-sensitive areas on the heatmaps. So maybe "identifying the approximate catchment location" is a better formulation than "catchment delineation" in the present state. We generally adapted the wording e.g. in Lines 54-46, Renamed section: "Spatial Input Sensitivity and Catchment Localization", 68, 264, 281f, 456

Primary purpose of this study was to show the usefulness of the 2D approach for discharge forecasting. Identifying the catchment location was a very interesting additional aspect, which seems to be worth investigating further in the future. We clarified some thoughts on how to move from our results to a delineation strategy (L473-490)

References

Frame, J., Kratzert, F., Klotz, D., Gauch, M., Shelev, G., Gilon, O., Qualls, L. M., Gupta, H. V., and Nearing, G. S.: Deep learning rainfall-runoff predictions of extreme events, Hydrol. Earth Syst. Sci. Discuss. [preprint], https://doi.org/10.5194/hess-2021-423, in review, 2021.

Gauch, M., Kratzert, F., Klotz, D., Nearing, G., Lin, J., and Hochreiter, S.: Rainfall–runoff prediction at multiple timescales with a single Long Short-Term Memory network. Hydrol. Earth Syst. Sci. https://doi.org/10.5194/hess-25-2045-2021, 2021.

Hsu, K., Gupta, H. V. and Sorooshian, S.: Artificial Neural Network Modeling of the RainfallRunoff Process, Water Resour. Res., 31(10), 2517–2530, doi:https://doi.org/10.1029/95WR01955, 1995.

Hussain, D., Hussain, T., Khan, A. A., Naqvi, S. A. A. and Jamil, A.: A deep learning approach for hydrological time-series prediction: A case study of Gilgit river basin, Earth Sci. Informatics, 13(3), 915–927, doi:10.1007/s12145-020-00477-2, 2020.

Kratzert, F., Klotz, D., Brenner, C., Schulz, K. and Herrnegger, M.: Rainfall–runoff modelling using Long Short-Term Memory (LSTM) networks, Hydrol. Earth Syst. Sci., 22(11), 6005–6022, doi:10.5194/hess-22-6005-2018, 2018.

Kratzert, F., Klotz, D., Herrnegger, M., Sampson, A. K., Hochreiter, S. and Nearing, G. S.: Toward Improved Predictions in Ungauged Basins: Exploiting the Power of Machine Learning, Water Resour. Res., 55(12), 11344–11354, doi:10.1029/2019WR026065, 2019a.

Kratzert, F., Klotz, D., Shalev, G., Klambauer, G., Hochreiter, S. and Nearing, G.: Towards learning universal, regional, and local hydrological behaviors via machine learning applied to large-sample datasets, Hydrol. Earth Syst. Sci., 23(12), 5089–5110, doi:http://dx.doi.org/10.5194/hess-23-5089-2019, 2019b.

Loffe, S. and Szegedy, C.: Batch Normalization: Accelerating Deep Network Training by Reducing Internal Covariate Shift. arXiv, 2015.

Maier, H. R. and Dandy, G. C.: The Use of Artificial Neural Networks for the Prediction of Water Quality Parameters, Water Resour. Res., 32(4), 1013–1022, doi:10.1029/96WR03529, 1996.

Maier, H. R. and Dandy, G. C.: Neural networks for the prediction and forecasting of water resources variables: a review of modelling issues and applications, Environ. Model. Softw., 15(1), 101–124, doi:https://doi.org/10.1016/S1364-8152(99)00007-9, 2000.

Srivastava, N., Hinton, G., Krizhevsky, A., Sutskever, I. and Salakhutdinov, R.: Dropout: A Simple Way to Prevent Neural Networks from Overfitting, J. Mach. Learn. Res., 15(1), 1929–1958, 2014.

Van, S. P., Le, H. M., Thanh, D. V., Dang, T. D., Loc, H. H. and Anh, D. T.: Deep learning convolutional neural network in rainfall–runoff modelling, J. Hydroinformatics, 22(3), 541–561, doi:10.2166/hydro.2020.095, 2020.

Zealand, C. M., Burn, D. H. and Simonovic, S. P.: Short term streamflow forecasting using artificial neural networks, J. Hydrol., 214(1), 32–48, doi:https://doi.org/10.1016/S0022-1694(98)00242-X, 1999

---

## Referee Report (RR1)

The revised manuscript is an improvement over the original. Some of the language used in asserting likelihood has been improved, the scope is more appropriate (e.g. delineation vs localization), and interesting new analyses have been performed. While details were added to the introduction, more are required as the background (especially on model interpretability) is still insufficient. Overall, the science is interesting but persistent imprecise and non-specific language remains as a substantial barrier.

The new analyses performed (varying the catchment location within the input domain) provides very exciting results. While many studies exist that build a machine learning model for streamflow prediction, I think that this sensitivity analysis makes the study stand out. Figure 7 shows very promising results and strongly supports the authors' case. I was very happy to see this. I think that this result is interesting, and it is worth the effort to improve the quality of the rest of the paper to continue towards publication. I think that this result will generate interest in the community.

Unfortunately, the writing is frequently imprecise and unclear. There were many instances where language surrounding precision was improved from the initial submission to this submission; however, many instances remain. This is a problem because vague phrases can be interpreted in multiple different ways, some of which are incorrect. These instances act as a barrier to readers' understanding. This is a substantial problem, but since the science seems to be sound and interesting, it is a problem that can (and should) be overcome. In this review I note many examples where improvements can be made, and why. It is my hope that the authors take the principles behind these comments and apply them throughout the text.

Comments:

Line 2: An ANN does not necessarily offer a "convenient solution" – perhaps "potential solution"?

Line 2: Input-output relationships are typically not "simple"

Line 4: "…few climate stations within a karst spring catchment are available" – it is not clear as to what an 'available climate station' means. Are there stations within these catchments but they are 'unavailable' for use? Or are they not located within these catchments? An alternative: e.g. "…few climate stations are located within or near karst spring catchments".

Line 4: "Hence spatial coverage… severe uncertainties." Introduce uncertainty into what?

Line 9: "based on" is not precise – the authors mean "use climate station data as input".

Line 10: The authors qualify their results as being "excellently suited to model karst spring discharge". This should be quantified with specific results (e.g. NSE or another quantitative

metric); otherwise, this abstract is not actually giving the reader the specific results of the study.

Line 11: This sentence is a bit hard to follow and can be made more clear if split in two, e.g. "The 2D-models show a better fit than the 1D-models in two of three cases and automatically learn to focus on the relevant areas of the input domain. By performing a spatial input sensitivity analysis...".

Line 17: "High heterogeneity" of what?

Line 19: Terms like "of them" are vague. What specifically does "them" refer to?

Line 19: "Certain level of background knowledge about the system" – All models, physically based or empirically based, require a 'certain level' of background knowledge (using an ANN still requires selecting input data/domain/etc). Do the authors mean that physically based models require more detailed system knowledge?

Line 20: "contrary" → should be "contrast"

Line 22: The sentence beginning with "Even though" seems to have two contradictory statements: first, that ANNs are not standard practice, and second, that they have been used for quite a long time. So what is it that the authors are trying to say here?

Line 23: The sentence beginning with "In fact" (and throughout the text) should be improved by using an active voice. E.g. "Johannet et al. (1994) showed that modelling water infiltration in karstic aquifers is possible with ANNs and was one of the first applications of ANNs in water related research."

Line 25: Sentence starting with "The number of applications" is not very clear (what does "even more accelerating" mean?). An alternative: "Applications of ANNs in hydrology have received a substantial amount of research attention, for example for rainfall-runoff modeling (e.g. Maier and Dandy (2000), Maier et al. (2010)), for groundwater applications (e.g. Rajaee et al. (2019)) and for hydrology and water resources in general (e.g. Sit et al. (2020))."

Line 28: Here, a transition from ANNs to CNNs should be included. What are the limitations of the fully-connected ANNs described in these reviews (e.g. MLP models)? Why not use them here? What are the characteristics of CNNs? Why are they suitable for application here? Then state that you use CNNs which wave been successfully applied etc etc. This will help improve the logical flow of the introduction: Karst systems are complicated; fully-connected ANNs are good for predictive modelling in hydrology; ANNs are limited by x y z; CNNs are beneficial because of a b c; so we use CNNs here to address the challenges in karst modelling.

Line 34: "Rudimentary" is a negative descriptor. Best to remove.

Line 35: "Especially… results," can be removed, unless the authors wish to get into conceptual models in more detail.

Line 39: "For this… catchment either." These two sentences have several examples of what I mean when I mention imprecision limiting reader understanding. "For this" – for what? "Reasonable time periods" – what is reasonable? "Various products" – like what, specifically? "Extract corresponding time series" – time series of what? "Knowledge of the spring catchment" – what kind of knowledge? "The available grid cells do not exactly match the catchment either" – match in what way? What does "exactly match" mean? The total lack of specificity greatly limits the informative content of these sentences.

Lines 42 – 49: These sentences are about catchment delineation. I think this could be a new paragraph since it does not really connect with the ideas of spatio-temporal completeness of the prior sentences. Since the goal of the paper has moved from delineation to localization, this should be reflected here as well; e.g. the ideas can flow from why delineation is a goal, how it is done, to why localization is a goal, to how it is done, to then how the authors propose to do it.

Line 54: "by themselves" – I think this should not be used as a descriptor for what the models are doing. Perhaps "automatically learn" is a more accurate way to say this.

Line 55: "… 2D-CNN processes … time series." This phrase is an example of when unclear language can imply an incorrect understanding. A reader could interpret this as meaning that the 2DCNN and 1DCNN are entirely independent of one another, when really they are not (e.g. both components are used for forecasting of the spring discharge time series). The 2D CNN learns spatial features and the 1D CNN learns the temporal relationship between those spatial features – perhaps that's what the authors mean here.

Line 64: I still have trouble with this phrasing. I think it is presumptuous to expect ANNs to be "superior". I think it is better to phrase as a potential advantage, e.g. "These requirements hold for our proposed methodology as well, but a potential advantage of ANNs is their nonlinearity which may better capture the nonlinear relationships between rainfall and discharge."

Introduction general comment: There should be information about deep learning model interpretability. It currently reads as though the authors found an interesting study and modified their method, without consideration for any other methods. There should be more information given (e.g. see studies like McGovern et al (2019), Fleming et al (2021a), and Fleming et al (2021b) and the references cited therein).

Line 79: What does "and except Unica mean"? Data was not easily accessible? Or that previous modelling approaches are not available for comparison?

Line 93: What does "to smaller parts" mean?

Line 124: What is meant here again by "parts"?  'from A and C (Fig. 1b)'

Line 124: "Clearly of the channel flow type" – Remove the word "clearly".

Line 164: "Including pumping data…" – Better to use active voice: "We do not use pumping data as input in this study because…"

Line 169: "They use a wide variation…": This is vague and doesn't offer any specific information. What are the input data used and what are the study goals?

Line 172: Why are these studies best compared to the authors' work?

Line 189: "Smallest" and "largest" are modifying "the relation of grid resolution to catchment size"; what relation?  The ratio?  Do the authors mean to say that Aubach is the smallest catchment in terms of area, or that it has the smallest ratio of grid cell area to catchment area?

Line 195: I don't understand what the authors are saying with "where the RADOLAN grid cell center lies in".  What does that mean?

Line 206: This claim can be made stronger by citing a study which quantifies the performance of at least the most important metric (precipitation).  There are many reanalysis products available.  Why did the authors choose the ones they chose?

Line 213: Provide references for CNNs applied for object recognition, image classification, and signal processing.

Line 218:  "Density of information" isn't a phrase that makes sense to me.

Line 222: As a reader, I am not going to trust statements like "from our experience in preliminary work".  Either provide results or do not include statements like this.

Line 224: "We have shown they are faster" -- seems to be in contrast to the statement in the response to referees, where the authors claimed that "there is no large difference in training time".  When the authors say they "have shown" – where?  Where is it shown that CNNs are faster than LSTMs?  The authors have made this claim but do not provide evidence.

Figure 2: Since the authors describe the input data difference on line 229, this Figure can be improved by making this distinction clear visually as well (e.g. label with weather station vs gridded reanalysis data input).

Line 231: See earlier comment about possible misinterpretation of this phrasing.

Line 232: I don't I agree with the statement that a 1D CNN instead of an LSTM improves the comparability between the two modelling approaches (2D-1D CNN vs 1D CNN).  In my opinion

those are just as comparable as a 2DCNN-LSTM vs LSTM.  I think this sentence and argument can be removed.

Line 261: "Sinus" – the authors mean "sine" or "sinusoid"

Line 272: "… physical meaning… others."  This phrase is confusing to me -- alternatively can say "CNN-LSTM models can learn to focus on specific areas of the spatially distributed input data".

Line 272: The sentence beginning with "We modify…" doesn't make sense ("... we demonstrate that this approach to qualitatively localize..."?)  I think some text was inadvertently deleted here.

Line 278: "Absolute parameter value of each pixel" --> "Absolute value of each variable at each pixel"

Line 282: Here, and throughout the remaining text, the authors refer to error but it is not clear what they mean.  How is error defined?  Difference between perturbed prediction and observation or between perturbed prediction and unperturbed prediction?

Line 282: "Repetitions" – I believe the authors mean "iterations".

Line 286: The sentence beginning with "Also these" is not easy to follow.  It should be rephrased to be more clear (e.g. "Temperature is spatially autocorrelated, meaning that temperature information from outside the catchment area may be used to infer temperature within the catchment area.  In contrast, precipitation is less spatially autocorrelated, meaning that precipitation information from outside the catchment area is less related to precipitation from inside the catchment area.  Therefore, we hypothesize that the within-catchment precipitation fields will be most important for the model's prediction, and we will test this hypothesis by visually inspecting the sensitivity maps produced by the modified approach of Anderson and Radic (2021).").

Line 301: "Daily" – I think the authors mean "diurnal".

Line 307: The authors claim that diurnal oscillations "no longer appear" in summer and autumn. They are still visible in Fig 3b.  They are diminished, but still appear.

Line 312: Here, and throughout the text, the authors refer to uncertainty but it is not clear what they mean.  Do they mean error between prediction and observation?  This should be clarified throughout the text.

Line 322: What range?  Instead of "same range" I think it is better to say "Our model has a similar, albeit slightly lower, NSE value compared to these three models.  One reason for this discrepancy could be that none of the previous studies…".

Line 338: The sentence starting with "Additionally" is unnecessarily complicated. It can be clarified by stating "The optimized model uses P, T, …".

Line 347: Sentence starting with "In total" – where do "we see" this? Where is it shown? And again, what is meant by "uncertainties in terms of input data"?

Line 368: It is more accurate to say "good" rather than "solid" (this occurs other times as well).

Line 368: What do the authors mean by "reaction"? Are they referring to the streamflow response (e.g. where observations increase strongly)?

Line 369: What is meant by "conservative"? An underestimation?

Line 374: What is meant by "even more true"? How can something be more true than observations?

Line 376 – 382: Could this conjecture not be easily verified by looking at the training predictions and seeing if these same errors persist? (E.g. underestimating spring peaks and overestimating low flows even before 2014)

Line 386: "by many orders of magnitude" – It looks like flow in Fig 2 varies by 2 orders of magnitude, not 'many'. Be specific.

Line 389: That low flows are no longer overestimated by very much seems to imply that the authors' previous claims about the impact of land use change may be incorrect

Line 392: What "same conceptual understanding" are the authors referring to?

Line 395: If there are some precipitation events in the gridded data that are not in the station data, wouldn't there be additional modelled discharge peaks in Fig 4b as compared to Fig 4a? The authors should be clearer in pointing to evidence to support their claim here.

Line 406 – 409: These studies should be moved to the introduction. Why are they in the discussion if they are not discussed? What did they do? What were the goals?

Line 419: Remove "easily" and provide a reference linking RH and radiation to evaporation.

Line 419: Remove "basically".

Line 425: Remove "definitely".

Line 426: Sentence beginning in "Though"– what does it mean? It is not clear.

Line 439: What does "knowledge extraction" mean? Be more specific.

Line 441: What does a "larger database" refer to?  More observations for training?  More input variables?  Be specific.

Line 454: "hardly the size of one grid cell" – just say "smaller than one grid cell".

Line 456: The phrase "main direction of the weather area" is not clear to me.  Upwind? Justify why it is the "main direction".  This phrase is used multiple times.

Line 458: "… this effect should be related to the size of the filter" – can the authors explain why and/or provide a reference for this?

Line 475: Remove "real world"

Line 476: Remove "even though… discharge signal."  It doesn't add to the discussion.

Line 491: Support point (i) with a reference and explain/clarify what is meant by "lower dampening" in point (ii).

Line 498: Given *what specifically* about the spatial resolution, heatmaps, and simulation results makes Unica springs the best example to investigate?  Make your thought process *explicit*, otherwise it sounds like the authors tried all three basins and are cherry-picking the results.

Line 501: "data frame" (and throughout the remaining text) – This term (and 'dataframe') is well used to refer to a type of data structure.  I believe the authors mean 'domain' or a similar term.

Line 504: What is meant by "extract the relevant input data" and where is this shown?  From the heat maps, or from the predicted discharge?

Line 537: It is not shown or explained how/why the 2D approach reduces the amount of work.

Line 543: What inaccuracies?  Be explicit and specific.

Line 545: "…we assume it can be used to delineate catchments quite accurately" – this is not a conclusion and is not really supported by the study in its current form.

Line 548:  Hard to conclude that 2D is overall superior due to the performance metrics -- maybe could state something like: "A key benefit of the 2D approach, which uses spatially discretized input data from climate reanalysis, is the spatially and temporally complete nature of the data and the number of variables available for study" or something to that effect.

Line 549: Sentence starting with "though": Increased effort for what as compared to what?

Table D1: Could the input sequence length be related to features observed in the predicted streamflow? E.g. diurnal oscillations are modelled in August seems it could be due to the model mapping temperature to flow, but due to the input time series length the model may not necessarily knowing if there is a snowpack available for melt (since this accumulated on longer timescales than are provided as input)? This point is worth considering and potentially adding to the discussion.

References

Fleming, S. W., Vesselinov, V. v, and Goodbody, A. G. (2021). Augmenting geophysical interpretation of data-driven operational water supply forecast modeling for a western US river using a hybrid machine learning approach. Journal of Hydrology 597, 126327. doi:https://doi.org/10.1016/j.jhydrol.2021.126327.

Fleming, S. W., Garen, D. C., Goodbody, A. G., McCarthy, C. S., and Landers, L. C. (2021). Assessing the new Natural Resources Conservation Service water supply forecast model for the American West: A challenging test of explainable, automated, ensemble artificial intelligence, 602, 126782, https://doi.org/https://doi.org/10.1016/j.jhydrol.2021.126782.

McGovern, A., Lagerquist, R., John Gagne, D., Jergensen, G. E., Elmore, K. L., Homeyer, C. R., et al. (2019). Making the Black Box More Transparent: Understanding the Physical Implications of Machine Learning. Bulletin of the American Meteorological Society 100, 2175–2199. doi:10.1175/BAMS-D-18-0195.1.

---

## Author Response (AR2)

Response Letter

We thank the referee for this extremely detailed textual analysis and the positive judgement of the manuscript in general. The propositions made by the referee considerably helped to improve the quality of the manuscript. We hence were able to improve the language in general and hope, that we are more precise throughout the text. Please find our answers in the following in red, while the review comments are reproduced verbatim in black. Line numbers in red, refer to the revised version of the manuscript. Some paragraphs were substantially rewritten, we therefore do not provide a Line number for every answer. The tracked-changes document hopefully helps out in these cases.

The revised manuscript is an improvement over the original. Some of the language used in asserting likelihood has been improved, the scope is more appropriate (e.g. delineation vs localization), and interesting new analyses have been performed. While details were added to the introduction, more are required as the background (especially on model interpretability) is still insufficient. Overall, the science is interesting but persistent imprecise and non-specific language remains as a substantial barrier.

The new analyses performed (varying the catchment location within the input domain) provides very exciting results. While many studies exist that build a machine learning model for streamflow prediction, I think that this sensitivity analysis makes the study stand out. Figure 7 shows very promising results and strongly supports the authors' case. I was very happy to see this. I think that this result is interesting, and it is worth the effort to improve the quality of the rest of the paper to continue towards publication. I think that this result will generate interest in the community.

Unfortunately, the writing is frequently imprecise and unclear. There were many instances where language surrounding precision was improved from the initial submission to this submission; however, many instances remain. This is a problem because vague phrases can be interpreted in multiple different ways, some of which are incorrect. These instances act as a barrier to readers' understanding. This is a substantial problem, but since the science seems to be sound and interesting, it is a problem that can (and should) be overcome. In this review I note many examples where improvements can be made, and why. It is my hope that the authors take the principles behind these comments and apply them throughout the text.

Comments:

Line 2: An ANN does not necessarily offer a "convenient solution" – perhaps "potential solution"?

We agree. We rephrased the respective part. Lines 1-3

Line 2: Input-output relationships are typically not "simple"

We rephrased this part, please see the answer above. Lines 1-3

Line 4: "…few climate stations within a karst spring catchment are available" – it is not clear as to what an 'available climate station' means. Are there stations within these catchments but they are 'unavailable' for use? Or are they not located within these catchments? An alternative: e.g. "…few climate stations are located within or near karst spring catchments".

We have adopted the suggestion. Thank you. Line 4

Line 4: "Hence spatial coverage… severe uncertainties." Introduce uncertainty into what?

Sentence revised to: "Hence, spatial coverage is often not satisfactory and can result in substantial uncertainties about the true conditions in the catchment, leading to lower model performance." Lines 5f.

Line 9: "based on" is not precise – the authors mean "use climate station data as input".

Correct, thank you. Sentence adapted to: "We compare the proposed approach both to existing modeling studies in these regions and to own 1D-CNN models that are conventionally trained with climate station input data." Lines 9-10

Line 10: The authors qualify their results as being "excellently suited to model karst spring discharge". This should be quantified with specific results (e.g. NSE or another quantitative metric); otherwise, this abstract is not actually giving the reader the specific results of the study.

Thank you for pointing out. We added NSE and KGE ranges to the Abstract. Line 11

Line 11: This sentence is a bit hard to follow and can be made more clear if split in two, e.g. "The 2D-models show a better fit than the 1D-models in two of three cases and automatically learn to focus on the relevant areas of the input domain. By performing a spatial input sensitivity analysis…".

Thank you very much. We have adopted the proposed modification as it stands. Lines 12-14

Line 17: "High heterogeneity" of what?

Karst systems in general are characterized by high structural heterogeneity due to the at least in large parts unknown conduit network, which controls the highly variable groundwater flow. We modified the respective sentence to clarify. Lines 17f.

Line 19: Terms like "of them" are vague. What specifically does "them" refer to?

We revised this section considerably. Please see Lines 18ff.

Line 19: "Certain level of background knowledge about the system" – All models, physically based or empirically based, require a 'certain level' of background knowledge (using an ANN still requires selecting input data/domain/etc). Do the authors mean that physically based models require more detailed system knowledge?

We revised this section considerably. See especially Line 22

Line 20: "contrary" à should be "contrast"

Thank you for pointing out. We revised our formulation, which is also now embedded in a considerably reworked paragraph. Line 21

Line 22: The sentence beginning with "Even though" seems to have two contradictory statements: first, that ANNs are not standard practice, and second, that they have been used for quite a long time. So what is it that the authors are trying to say here?

In our opinion this is indeed contradictory. Being a useful tool (stated before) and at the same time being used for a long time already (stated here), implies that ANNs may be an established approach. This is not the case. We therefore stick to this specific formulation.

Line 23: The sentence beginning with "In fact" (and throughout the text) should be improved by using an active voice. E.g. "Johannet et al. (1994) showed that modelling water infiltration in karstic aquifers is possible with ANNs and was one of the first applications of ANNs in water related research."

Done. Thank you for pointing out. Line 25f and throughout the manuscript.

Line 25: Sentence starting with "The number of applications" is not very clear (what does "even more accelerating" mean?). An alternative: "Applications of ANNs in hydrology have received a substantial amount of research attention, for example for rainfall-runoff modeling (e.g. Maier and Dandy (2000),

Maier et al. (2010)), for groundwater applications (e.g. Rajaee et al. (2019)) and for hydrology and water resources in general (e.g. Sit et al. (2020)).”

Thanks for pointing out. We restructured these sentences and added more background information as requested below. Lines 26ff.

Line 28: Here, a transition from ANNs to CNNs should be included. What are the limitations of the fully-connected ANNs described in these reviews (e.g. MLP models)? Why not use them here? What are the characteristics of CNNs? Why are they suitable for application here? Then state that you use CNNs which wave been successfully applied etc etc. This will help improve the logical flow of the introduction: Karst systems are complicated; fully-connected ANNs are good for predictive modelling in hydrology; ANNs are limited by x y z; CNNs are beneficial because of a b c; so we use CNNs here to address the challenges in karst modelling.

Done. Thank you for pointing out. See roughly Lines 25 to 44

Line 34: “Rudimentary” is a negative descriptor. Best to remove.

Done.

Line 35: “Especially… results,” can be removed, unless the authors wish to get into conceptual models in more detail.

Done.

Line 39: “For this… catchment either.” These two sentences have several examples of what I mean when I mention imprecision limiting reader understanding. “For this” – for what? “Reasonable time periods” – what is reasonable? “Various products” – like what, specifically? “Extract corresponding time series” – time series of what? “Knowledge of the spring catchment” – what kind of knowledge? “The available grid cells do not exactly match the catchment either” – match in what way? What does “exactly match” mean? The total lack of specificity greatly limits the informative content of these sentences.

Thank you for pointing out this specific example. It certainly helped to get your point. We revised this sentence and others throughout the text. e.g. Lines 65-68

Lines 42 – 49: These sentences are about catchment delineation. I think this could be a new paragraph since it does not really connect with the ideas of spatio-temporal completeness of the prior sentences. Since the goal of the paper has moved from delineation to localization, this should be reflected here as well; e.g. the ideas can flow from why delineation is a goal, how it is done, to why localization is a goal, to how it is done, to then how the authors propose to do it.

Thanks for pointing out. We have revised this section. Lines 74-84

Line 54: “by themselves” – I think this should not be used as a descriptor for what the models are doing. Perhaps “automatically learn” is a more accurate way to say this.

Changed throughout the manuscript, thank you.

Line 55: “… 2D-CNN processes … time series.” This phrase is an example of when unclear language can imply an incorrect understanding. A reader could interpret this as meaning that the 2DCNN and 1DCNN are entirely independent of one another, when really they are not (e.g. both components are used for forecasting of the spring discharge time series). The 2D CNN learns spatial features and the 1D CNN learns the temporal relationship between those spatial features – perhaps that's what the authors mean here.

Again, thanks for pointing out. We rephrased these (and other related) sentences to become clearer. See e.g. lines 85ff or section 3.2

Line 64: I still have trouble with this phrasing. I think it is presumptuous to expect ANNs to be "superior". I think it is better to phrase as a potential advantage, e.g. "These requirements hold for our proposed methodology as well, but a potential advantage of ANNs is their nonlinearity which may better capture the nonlinear relationships between rainfall and discharge."

Thank you again for your effort to propose adequate changes. We adapt this one to our text verbatim. Lines 99-100

Introduction general comment: There should be information about deep learning model interpretability. It currently reads as though the authors found an interesting study and modified their method, without consideration for any other methods. There should be more information given (e.g. see studies like McGovern et al (2019), Fleming et al (2021a), and Fleming et al (2021b) and the references cited therein).

Thank you for improving also our storyline. We added a paragraph on explainable AI and model interpretability to the introduction. Lines 45ff.

Line 79: What does "and except Unica mean"? Data was not easily accessible? Or that previous modelling approaches are not available for comparison?

We made the reference clearer and revised the respective sentences. Esp. L 121f. but generally lines 101 ff.

Line 93: What does "to smaller parts" mean?

We change the wording ("proportions") and hope it gets clearer now.  L150

Line 124: What is meant here again by "parts"? 'from A and C (Fig. 1b)'

Parts as indicated on the map. We now write "regions".  L182

Line 124: "Clearly of the channel flow type" – Remove the word "clearly".

Done. L182

Line 164: "Including pumping data…" – Better to use active voice: "We do not use pumping data as input in this study because…"

Done. L222

Line 169: "They use a wide variation…": This is vague and doesn't offer any specific information. What are the input data used and what are the study goals?

We revised the literature discussion throughout the manuscript. Thus, this paragraph was completely rephrased and moved to the Introduction section Lines 101ff

Line 172: Why are these studies best compared to the authors' work?

We revised the literature discussion throughout the manuscript, and now better explain these aspects. The paragraph was completely rephrased and moved to the Introduction section. Lines 101ff

Line 189: "Smallest" and "largest" are modifying "the relation of grid resolution to catchment size"; what relation? The ratio? Do the authors mean to say that Aubach is the smallest catchment in terms of area, or that it has the smallest ratio of grid cell area to catchment area?

We revised the wording in the manuscript and now speak of "ratio", however this specific sentence was removed during rewriting the corresponding section.

Line 195: I don't understand what the authors are saying with "where the RADOLAN grid cell center lies in". What does that mean?

We rephrased it to: "The higher resolved precipitation data from RADOLAN is thus augmented with climate parameter values from ERA5-Land, which were downscaled and re-gridded to match the 1x1 km RADOLAN grid."

We hope this answers the question. L245f.

Line 206: This claim can be made stronger by citing a study which quantifies the performance of at least the most important metric (precipitation). There are many reanalysis products available. Why did the authors choose the ones they chose?

We added a study that evaluates precipitation for both data sets and we better explain why we chose them. L259

Line 213: Provide references for CNNs applied for object recognition, image classification, and signal processing.

Done. L266f.

Line 218: "Density of information" isn't a phrase that makes sense to me.

We have revised and augmented this passage: "The latter performs down-sampling of the produced feature maps and summarizes the features detected in the input. This decreases the total number of parameters of the model and makes it approximately invariant to small translations of the input (Goodfellow et al. 20116)" Lines 271f

Line 222: As a reader, I am not going to trust statements like "from our experience in preliminary work". Either provide results or do not include statements like this.

We removed this part of the respective statement and now only refer to the cited studies.

Line 224: "We have shown they are faster" -- seems to be in contrast to the statement in the response to referees, where the authors claimed that "there is no large difference in training time". When the authors say they "have shown" – where? Where is it shown that CNNs are faster than LSTMs? The authors have made this claim but do not provide evidence.

After stating that CNNs are faster we now cite Wunsch et al. (2021), where we investigated this aspect in the context of groundwater level forecasting and showed that CNNs are systematically faster than LSTMs. In our last response to referees, we referred to the difference between 2DCNN-1DCNN and 2DCNN-LSTM, whereas we here speak of 1D-CNNs. We have additionally reworked the whole paragraph (including additional references), to point out that the speed is only one aspect in our choice for CNN instead of LSTM models. Lines 277ff.

Figure 2: Since the authors describe the input data difference on line 229, this Figure can be improved by making this distinction clear visually as well (e.g. label with weather station vs gridded reanalysis data input).

Thank you for this proposition. We modified the Figure accordingly.

Line 231: See earlier comment about possible misinterpretation of this phrasing.

We similarly rewrote this whole paragraph. See also our answer to Line 224

Line 232: I don't I agree with the statement that a 1D CNN instead of an LSTM improves the comparability between the two modelling approaches (2D-1D CNN vs 1D CNN). In my opinion those are just as comparable as a 2DCNN-LSTM vs LSTM. I think this sentence and argument can be removed.

We think that using 2DCNN-1DCNN instead of 2DCNN-LSTM allow better conclusions on the importance of the input data and its influence on the modeling result. Using a LSTM would mix the data influence with the question whether the model performance changed due to the LSTM. We elaborate this aspect now in the manuscript. Thanks for pointing out. Lines 287-289

Line 261: "Sinus" – the authors mean "sine" or "sinusoid"

Yes, thank you. A translation error from our side.

Line 272: "… physical meaning… others." This phrase is confusing to me -- alternatively can say "CNN-LSTM models can learn to focus on specific areas of the spatially distributed input data".

We adapted the proposed change. Thank you.  Lines 325f.

Line 272: The sentence beginning with "We modify…" doesn't make sense ("... we demonstrate that this approach to qualitatively localize..."?) I think some text was inadvertently deleted here.

You are right, thanks for noticing. We now go with: "We modify this approach and transfer it to karst spring modeling, where we demonstrate that this approach is suited to qualitatively localize karst catchment locations.  "Lines 326f.

Line 278: "Absolute parameter value of each pixel" --> "Absolute value of each variable at each pixel"

We adapted the proposed change. Thank you. L330

Line 282: Here, and throughout the remaining text, the authors refer to error but it is not clear what they mean. How is error defined? Difference between perturbed prediction and observation or between perturbed prediction and unperturbed prediction?

It's the difference between the perturbed prediction and the unperturbed prediction. Thank you for pointing out this missing detail, which we now mention in the text. Lines 334-335

Line 282: "Repetitions" – I believe the authors mean "iterations".

Corrected, thank you.

Line 286: The sentence beginning with "Also these" is not easy to follow. It should be rephrased to be more clear (e.g. "Temperature is spatially autocorrelated, meaning that temperature information from outside the catchment area may be used to infer temperature within the catchment area. In contrast, precipitation is less spatially autocorrelated, meaning that precipitation information from outside the catchment area is less related to precipitation from inside the catchment area. Therefore, we hypothesize that the within-catchment precipitation fields will be most important for the model's prediction, and we will test this hypothesis by visually inspecting the sensitivity maps produced by the modified approach of Anderson and Radic (2021).").

Thank you again for your excellent propositions. We rephrased these sentences accordingly. Lines 336ff.

Line 301: "Daily" – I think the authors mean "diurnal".

Yes, thank you. We changed several occurrences throughout the text.

Line 307: The authors claim that diurnal oscillations "no longer appear" in summer and autumn. They are still visible in Fig 3b. They are diminished, but still appear.

You are correct. We changed it to: "now are diminished" L. 358

Line 312: Here, and throughout the text, the authors refer to uncertainty but it is not clear what they mean. Do they mean error between prediction and observation? This should be clarified throughout the text.

We changed these occurrences throughout the manuscript. E.g. here we now use "error source". L361f. Sie also our answer to "Line 347"

Line 322: What range? Instead of "same range" I think it is better to say "Our model has a similar, albeit slightly lower, NSE value compared to these three models. One reason for this discrepancy could be that none of the previous studies…".

Thank you for your excellent proposition. This sentence is, however, removed due to restructuring and rewriting the section. Lines 363-369

Line 338: The sentence starting with "Additionally" is unnecessarily complicated. It can be clarified by stating "The optimized model uses P, T, …".

Thank you, we simplified the sentence to: "The optimized model uses P, T, Tsin, SMLT, SF, SWVL1/2/4 as inputs, thus omits E and SWVL3." L.371

Line 347: Sentence starting with "In total" – where do "we see" this? Where is it shown? And again, what is meant by "uncertainties in terms of input data"?

We rephrased this to: "In total, we think that both the 1D and the 2D-approach for this catchment bear substantial shortcomings in terms of how well the input data represents the true conditions in the catchment, even though the simulation results are generally very accurate" Lines: 392-394

Line 368: It is more accurate to say "good" rather than "solid" (this occurs other times as well).

Thank you. We changed several occurrences in the text. L404

Line 368: What do the authors mean by "reaction"? Are they referring to the streamflow response (e.g. where observations increase strongly)?

Yes, thank you for pointing out. "Response" is the accurate wording. L405

Line 369: What is meant by "conservative"? An underestimation?

Probably another mistake made inadvertently by transposing an additional meaning of a word into another language. Sorry for that. What we meant is that the slope is not as steep as for the observed recession. We rephrased the two occurrences in the text to be more precise. L 405 ff.

Line 374: What is meant by "even more true"? How can something be more true than observations?

As we explain in the two sentences before, under certain conditions it is "impossible to accurately monitor the inflow conditions" This means at times of the plateau like peaks, we do not know the true conditions. Therefore, the simulated conditions might be conceptually true, thus representing a flow

variation we cannot observe. Nevertheless, we rephrased these sentences, to improve clarity. Lines 408ff

Line 376 – 382: Could this conjecture not be easily verified by looking at the training predictions and seeing if these same errors persist? (E.g. underestimating spring peaks and overestimating low flows even before 2014)

Good point, thanks for pointing out. We think it is not that easy. As you can see in Table 1, we split the time series into four parts: 1981-2012 for Training, 2013+2014 for Validation (Early Stopping), 2015+2016 for HP Optimization and finally 2017+2018 for testing. The considered period of the environmental changes 2014-2018 is not part of the training data, but nevertheless has influence on the modeling process, since it is covered by the period used for HP optimization and early stopping. So even by looking at the different parts of the data it should be hard to disentangle these effects. Anyhow, we found it worth to notice. We rephrased this part: Lines 414ff.

Line 386: "by many orders of magnitude" – It looks like flow in Fig 2 varies by 2 orders of magnitude, not 'many'. Be specific.

Corrected. Thank you. L427

Line 389: That low flows are no longer overestimated by very much seems to imply that the authors' previous claims about the impact of land use change may be incorrect Line 392: What "same conceptual understanding" are the authors referring to?

Thank you for pointing out. We admit that we missed this flaw in our argumentation. We thus rephrased both parts, still mention the environmental change but do not hold it accountable for the 1D model performance during low flow. We also explain what we mean with same conceptual understanding. Lines 413ff. and Lines 433ff.

Line 395: If there are some precipitation events in the gridded data that are not in the station data, wouldn't there be additional modelled discharge peaks in Fig 4b as compared to Fig 4a? The authors should be clearer in pointing to evidence to support their claim here.

You are right, we rewrote this sentence. L 436f.

Line 406 – 409: These studies should be moved to the introduction. Why are they in the discussion if they are not discussed? What did they do? What were the goals?

We moved these studies to the introduction. Thank you for pointing out. L. 106-107

Line 419: Remove "easily" and provide a reference linking RH and radiation to evaporation.

Done. L45ff.

Line 419: Remove "basically".

Done.

Line 425: Remove "definitely".

Done.

Line 426: Sentence beginning in "Though"– what does it mean? It is not clear.

We rephrased this section. L462ff.

Line 439: What does "knowledge extraction" mean? Be more specific.

Line 441: What does a "larger database" refer to? More observations for training? More input variables? Be specific.

We added some details and moved this part to the introduction. L109ff.

Line 454: "hardly the size of one grid cell" – just say "smaller than one grid cell".

Done. Thank you. L479

Line 456: The phrase "main direction of the weather area" is not clear to me. Upwind? Justify why it is the "main direction". This phrase is used multiple times.

Sorry for this imprecise description, presumably resulting from a translation problem. In German this literally describes the direction at a certain location from which weather phenomena usually originate most of the time. For example, when precipitation events usually come from the West, then this is what we mean. We acknowledge that this obviously is confusing. We decided to remove these statements from the text.

Line 458: "… this effect should be related to the size of the filter" – can the authors explain why and/or provide a reference for this?

We rephrased this statement. Lines 482-486

Line 475: Remove "real world"

Done, thank you.

Line 476: Remove "even though… discharge signal." It doesn't add to the discussion.

Done.

Line 491: Support point (i) with a reference and explain/clarify what is meant by "lower dampening" in point (ii).

We have removed this statement from our text.

Line 498: Given *what specifically* about the spatial resolution, heatmaps, and simulation results makes Unica springs the best example to investigate? Make your thought process *explicit*, otherwise it sounds like the authors tried all three basins and are cherry-picking the results.

Done. L520ff

Line 501: "data frame" (and throughout the remaining text) – This term (and 'dataframe') is well used to refer to a type of data structure. I believe the authors mean 'domain' or a similar term.

Dataframe is also well used in GIS context to the frame on a map with two-dimensional content displayed. Nevertheless, to avoid misunderstandings, we have changed the wording throughout the text to "considered area of the input data"

Line 504: What is meant by "extract the relevant input data" and where is this shown? From the heat maps, or from the predicted discharge?

By "relevant input data" we mean "relevant grid cells within the considered input area". We changed the wording accordingly. L527ff.

Line 537: It is not shown or explained how/why the 2D approach reduces the amount of work.

We do explain it now. Thanks for pointing out. L562ff.

Line 543: What inaccuracies? Be explicit and specific.

We rephrased the whole section. L566ff.

Line 545: "…we assume it can be used to delineate catchments quite accurately" – this is not a conclusion and is not really supported by the study in its current form.

We rephrased the whole section. L 572ff.

Line 548: Hard to conclude that 2D is overall superior due to the performance metrics – maybe could state something like: "A key benefit of the 2D approach, which uses spatially discretized input data from climate reanalysis, is the spatially and temporally complete nature of the data and the number of variables available for study" or something to that effect.

We rephrased the respective sentences to make our reasoning clearer and also added the proposed sentence. Thank you. L574ff.

Line 549: Sentence starting with "though": Increased effort for what as compared to what?

We rephrased this sentence. L578f.

Table D1: Could the input sequence length be related to features observed in the predicted streamflow? E.g. diurnal oscillations are modelled in August seems it could be due to the model mapping temperature to flow, but due to the input time series length the model may not necessarily be knowing if there is a snowpack available for melt (since this accumulated on longer timescales than are provided as input)? This point is worth considering and potentially adding to the discussion.

Thank you for this interesting remark. We provided Tsin as input variable to cope with this aspect. Using this sine signal input, the model (presumably) learns the season and the current position in the annual cycle. It may theoretically therefore be aware of a potential snow pack. You state that it might be due to the model mapping temperature to flow – we think that of course T is (somehow) mapped to flow, otherwise the model would not use it as an input variable. The question is rather if it was done in the correct way, such as deriving maybe seasonality from it, reducing flow in periods of high evapotranspiration and so on. To explore this aspect is an interesting idea, using explainable AI methods (e.g. SHAP values), to explore the influence of each input on the model output. This is however, beyond the scope of our study.

References

Fleming, S. W., Vesselinov, V. v, and Goodbody, A. G. (2021). Augmenting geophysical interpretation of data-driven operational water supply forecast modeling for a western US river using a hybrid machine learning approach. Journal of Hydrology 597, 126327. doi: https://doi.org/10.1016/j.jhydrol.2021.126327.

Fleming, S. W., Garen, D. C., Goodbody, A. G., McCarthy, C. S., and Landers, L. C. (2021). Assessing the new Natural Resources Conservation Service water supply forecast model for the American West: A challenging test of explainable, automated, ensemble artificial intelligence, 602, 126782, https://doi.org/10.1016/j.jhydrol.2021.126782.

McGovern, A., Lagerquist, R., John Gagne, D., Jergensen, G. E., Elmore, K. L., Homeyer, C. R., et al. (2019). Making the Black Box More Transparent: Understanding the Physical Implications of Machine Learning. Bulletin of the American Meteorological Society 100, 2175–2199. doi: https://doi.org/10.1175/BAMS-D-18-0195.1.